# Time Discretization-Invariant
# Safe Action Repetition for Policy Gradient Methods

**Seohong Park**
Seoul National University
artberryx@snu.ac.kr

**Jaekyeom Kim**
Seoul National University
jaekyeom@snu.ac.kr

**Gunhee Kim**
Seoul National University
gunhee@snu.ac.kr

## Abstract

In reinforcement learning, continuous time is often discretized by a time scale $\delta$, to which the resulting performance is known to be highly sensitive. In this work, we seek to find a $\delta$-*invariant* algorithm for policy gradient (PG) methods, which performs well regardless of the value of $\delta$. We first identify the underlying reasons that cause PG methods to fail as $\delta \to 0$, proving that the variance of the PG estimator can diverge to infinity in stochastic environments under a certain assumption of stochasticity. While durative actions or action repetition can be employed to have $\delta$-invariance, previous action repetition methods cannot immediately react to unexpected situations in stochastic environments. We thus propose a novel $\delta$-invariant method named *Safe Action Repetition (SAR)* applicable to any existing PG algorithm. SAR can handle the stochasticity of environments by adaptively reacting to changes in states during action repetition. We empirically show that our method is not only $\delta$-invariant but also robust to stochasticity, outperforming previous $\delta$-invariant approaches on eight MuJoCo environments with both deterministic and stochastic settings. Our code is available at https://vision.snu.ac.kr/projects/sar.

## 1 Introduction

Deep reinforcement learning (RL) has demonstrated phenomenal achievements in a wide array of tasks, including superhuman game-playing [19, 27] and controlling complex robots [8, 13]. Most RL algorithms are based on an Markov Decision Process (MDP), which is a discrete-time control process for the iteration of observing a state and performing an action. However, numerous real-world problems such as robotic manipulation and autonomous driving are defined in continuous time, which does not directly fit the MDP setting. To fill this gap, continuous time is often discretized by a *discretization time scale* $\delta$, where the RL agent makes a decision at every $\delta$. It has been shown that RL algorithms are greatly sensitive to this hyperparameter [1, 36]. For instance, altering $\delta$ via frame skipping leads to drastic performance differences [1, 4]. Indeed, an excessively high $\delta$ precludes the agent from making fine-grained decisions, which is likely to cause performance degradation.

On average, the agent could perform equally well or better with a lower $\delta$ than with a higher $\delta$, since the agent can make decisions more frequently. However, Baird [2] and Tallec et al. [36] theoretically proved that the standard Q-learning fails when $\delta \to 0$ as the action-value (Q) function collapses to the state-value (V) function, eliminating preferences between actions. As will be shown in Section 4.1, policy gradient (PG) methods fail as well with an infinitesimal $\delta$ for the following three reasons: (1) The variance of the gradient estimator explodes. (2) Exploration ranges may become highly limited. (3) Infinitely many decision steps are required. The latter two also apply to Q-learning methods.

Therefore, it is generally required to differently set an appropriate $\delta$ for each continuous environment. Indeed, continuous control environments in MuJoCo [37] have different discretization time scales from one another, ranging from 0.008s (Hopper) to 0.05s (InvertedDoublePendulum). However, such

tuning of $\delta$ could be burdensome when applying RL algorithms to new environments, considering its significant influence on performance [1, 36]. Furthermore, even if the optimal $\delta$ is found in simulation, trained policies may not be transferable to real-world settings, since physical sensors often have their unique sampling frequencies.

There have been proposed some methods for robustness to discretization of time scales. $\delta$-invariant methods could bring several advantages: (1) It obviates the need for tuning $\delta$ on each continuous control environment. (2) They can achieve better performance by utilizing more fine-grained control with a low $\delta$. (3) Using a policy with adaptive decision frequencies (as a variant of $\delta$-invariant policies), the agent can efficiently take actions only when necessary, which could expedite training without losing agility. Tallec et al. [36] introduced an algorithm based on Advantage Updating [2], which can make existing Q-learning methods (*e.g.*, DQN [18] and DDPG [16]) invariant to $\delta$ by preventing them from Q-function collapse. For PG methods, Munos [21] and Wawrzynski [38] proposed methods that can cope with fine time discretization. However, these methods either assume access to the gradient of the reward function or require an infinite number of decision steps (or training steps) when $\delta \to 0$, both of which could hinder its application to real-world environments.

We aim at proposing an efficient $\delta$-invariant approach applicable to existing PG methods such as PPO [30], TRPO [28] and A2C [20]. One straightforward approach may be to take *durative actions* by making policies produce both *actions* and their *durations*. Such an approach is practically equivalent to prior work on *action repetition* whose policies output both actions and the number of action repetitions [15, 31], since continuous control environments such as MuJoCo often already provide discretized time scales. However, prior approaches to action repetition possess some limitations. For example, there is no way to stop repeating a chosen action during a repetition period, which means that they are not capable of immediately handling unexpected events in stochastic environments. This could lead to catastrophic failure in some real-world settings such as autonomous driving.

We thus propose an alternative approach named *Safe Action Repetition* (SAR) with the key idea of repeating an action until the agent exits its *safe region*. Our policy produces both an *action* and a *safe region* in the state space, only within which the chosen action is repeated. SAR enables any PG algorithm to not only be $\delta$-invariant but also be robust to stochasticity such as unexpected events in the environment, because such situations lead the agent's state to be outside of the safe region, immediately causing the cease of the current action. We apply the proposed method to several PG algorithms and empirically show that SAR indeed exhibits $\delta$-invariance on various MuJoCo environments and outperforms baselines on both deterministic and stochastic settings.

Our contributions can be summarized as follows:

- We first provide a more general proof on the variance explosion of the PG estimator, which is the main reason why PG methods fail as $\delta \to 0$. We then show that temporally extended actions can resolve the failure mode of PG algorithms with a low $\delta$.

- We introduce a novel $\delta$-invariant method named SAR applicable to any PG method on continuous control domains. To the best of our knowledge, this is the first action repetition (or durative action) method that repeats an action based on the agent's *state*, rather than a precomputed action duration. As a result, SAR can cope with unexpected situations in stochastic environments, which existing action repetition methods cannot handle.

- We apply SAR to three PG methods, PPO, TRPO and A2C, and empirically demonstrate that our SAR method is mostly invariant to $\delta$ on eight MuJoCo environments. We also verify its robustness to stochasticity via three different stochastic settings on each MuJoCo environment. Our method also outperforms previous $\delta$-invariant approaches such as FiGAR-C [31] and DAU [36] on those settings.

## 2 Related Work

**Continuous-time RL.** Reinforcement learning in continuous-time domains has long been studied with various approaches [2, 3, 5, 6, 7, 21, 22]. Bradtke and Duff [3] extended existing Q-learning and temporal difference methods to semi-MDPs, which can be viewed as a continuous-time generalization of MDPs. Doya [5] developed a continuous actor-critic method based on the Hamilton-Jacobi-Bellman (HJB) equation, a continuous-time counterpart of the Bellman equation, approximating policies and value functions with radial basis functions.

**Time discretization.** Another line of research to cope with continuous-time environments is to use finely discretized MDPs. Baird [2] first informally presented that when $\delta \to 0$, two different Q values on the same state will eventually collapse to the V value, such that $Q(s, a_1) \approx Q(s, a_2) \approx V(s)$, since the influence of each action is vanished by the infinitesimal time scale. They proposed *Advantage Updating* as a solution to avoid the collapse by appropriately scaling the advantage function $A(s, a) = \frac{Q(s,a) - V(s)}{\delta}$. More recently, Tallec et al. [36] theoretically proved the existence of collapse in low-$\delta$ settings and extended Advantage Updating for deep neural networks, showing its $\delta$-invariance on classic control benchmarks. For PG methods, Munos [21] first demonstrated that the variance of the policy gradient estimate can be infinite when $\delta \to 0$, and proposed an algorithm based on pathwise derivatives, in which the variance of the estimator decreases to 0 when $\delta \to 0$. However, it assumes that the gradient of the reward function $\nabla r(s, a)$ is known to the agent. Korenkevych et al. [14], Wawrzynski [38] proposed methods based on autocorrelated noise that could prevent the variance explosion. Notably, all of these approaches choose an action at every $\delta$. However, this makes it infeasible to train when $\delta$ is nearly zero as the number of decision steps goes to infinity. In our work, we employ durative actions to achieve $\delta$-invariance on PG methods, which could resolve the problem of variance explosion and infinite decision steps.

**Action repetition.** Our proposed method is closely related to existing action repetition methods. Most works on action repetition let the policy also determine how many times (or how long) an action is repeated. DFDQN [15] doubled the action space by mapping a half of it to actions repeated $r_1$ times and the other half to those repeated $r_2$ times, where $r_1$ and $r_2$ are fixed hyperparameters. FiGAR [31] introduced a repetition policy $\pi(x|s)$ in addition to the original action policy $\pi(a|s)$ so that the agent repeats the action $x$ times, where $x$ is selected from a predefined set $W$; *e.g.*, $W = \{1, 2, \dots, 30\}$. Metelli et al. [17] theoretically analyzed the performance of the optimal policy when a fixed action repetition count is given, and proposed a heuristic to approximately choose the optimal control frequency. On the other hand, we take a completely different approach where our policy produces not repetition counts (or action durations) but *safe regions*, which enables the agent to adaptively stop action repetition when facing unexpected events in stochastic environments.

Finally, action repetition is related to the options framework [34] in that they both use temporally extended actions. In the options framework, the agent learns both a high-level inter-option policy and a low-level intra-option policy, where action repetition can be interpreted as a special case of an intra-option policy. However, one crucial difference between them is that within the options framework, the agent has to produce each $\delta$-discretized low-level action (even with open-loop options), which makes having $\delta$-invariance non-trivial, unlike the action repetition approach.

## 3 Preliminaries

We consider a continuous-time MDP $\mathcal{M} = (\mathcal{S}, \mathcal{A}, r, F, \gamma)$ [3, 5], where $\mathcal{S}$ is a continuous state space, $\mathcal{A}$ is a bounded action space, $F \colon \mathcal{S} \times \mathcal{A} \to \mathcal{S}$ is a transition dynamics function, $r \colon \mathcal{S} \times \mathcal{A} \to \mathbb{R}$ is a reward function and $\gamma \in (0, 1]$ is a discount factor. For simplicity, we assume the environment has deterministic transition dynamics; we refer to Munos and Bourgine [22] for the stochastic case involving Brownian motion. The transition dynamics and the return $R(\tau)$ are given by

$$s(t) = s(0) + \int_0^t F(s(t'), a(t'))dt', \quad R(\tau) = \int_0^\infty \gamma^{t'} r(s(t'), a(t'))dt', \quad (1)$$

where $s(t)$ and $a(t)$ respectively denote the state and action at time $t$, and $\tau$ denotes the whole trajectory consisting of states and actions.

Following Tallec et al. [36], we define a discretized version of $\mathcal{M}$ as $\mathcal{M}_\delta = (\mathcal{S}, \mathcal{A}, r_\delta, F_\delta, \gamma_\delta)$ with a *discretization time scale* $\delta > 0$, where the agent observes a state and performs an action at every $\delta$. We respectively denote the state and action at $i$-th step as $s_i$ and $a_i$, where $s_i$ in $\mathcal{M}_\delta$ corresponds to $s(i\delta)$ in $\mathcal{M}$ and $a_i$ is maintained during the time interval $[i\delta, (i + 1)\delta)$. In $s(t)$, $t$ indicates the *physical* time, and $i$ in $s_i$ is the number of steps taken. The reward $r_i$ and return $R_\delta(\tau)$ are defined as

$$r_i = r_\delta(s_i, a_i) = r(s(i\delta), a(i\delta))\delta, \quad R_\delta(\tau) = \sum_{i=0}^\infty \gamma_\delta^i r_i, \quad (2)$$

where the discount factor is $\gamma_\delta = \gamma^\delta$. Given a deterministic policy $\pi \colon \mathcal{S} \to \mathcal{A}$, the continuous value function $V^\pi(s)$ and the discretized one $V_\delta^\pi$ are defined as $V^\pi(s) = \mathbb{E}_{\tau \sim p_\pi(\tau)}[R(\tau)|s(0) = s]$ and

$V_\delta^\pi(s) = \mathbb{E}_{\tau \sim p_\pi(\tau)}[R_\delta(\tau)|s_0 = s]$, respectively. Tallec et al. [36] proved that $V_\delta^\pi$ converges to $V^\pi$ when $\delta \to 0$ under smoothness assumptions.

In the rest of the paper, we will focus on $\mathcal{M}_\delta$ (*i.e.*, $\mathcal{M}$ with time discretization) and omit the subscript $\delta$ unless it is necessary. Also, as in ordinary discrete-time MDPs [35], we consider stochastic transition dynamics $p(s_{i+1}|s_i, a_i)$ and stochastic policy $\pi(a_i|s_i)$ instead of deterministic ones.

## 4 Safe Action Repetition

We first show that $\delta \to 0$ leads to failure in policy gradient (PG) methods (Section 4.1), and propose that durative actions or action repetition can be a solution to the failure mode. We then point out that existing action repetition methods have drawbacks in the presence of unexpected events in stochastic environments (Section 4.2). As a solution, we propose *Safe Action Repetition* (SAR) as a novel $\delta$-invariant approach for PG algorithms, which is robust to such stochasticity (Section 4.3). We additionally suggest a variant of SAR to deal with non-Markovian environments (Section 4.4).

### 4.1 Policy Gradients with Infinitesimal Discretization of Time Scale

A smaller discretization time scale should not lead to worse maximum returns on average, since it allows the agent to perform more fine-grained actions. However, both Q-learning and PG methods are subjected to fail with a too small $\delta$. We introduce three reasons why PG methods fail. We refer to Baird [2], Tallec et al. [36] for further discussion on Q-learning methods.

**Variance explosion of the policy gradient estimator.** We show that the variance of the PG estimator can diverge to infinity with a decrease in $\delta$. While this was first shown in Munos [21] via a simple illustrative example, we provide a more general proof without assuming a particular environment. Specifically, we prove that the variance explosion problem can arise in any stochastic environment where the variance of its return conditioned on actions is greater than a small positive constant.

Let us consider a stochastic environment that has a finite physical time limit $T$ and a discretization time scale $\delta$. It follows that the number of decision steps (or actions) in a single rollout is $N = T/\delta$. Let the policy $\pi_\theta(a_i|s_i)$ be parameterized by $\theta$, the distribution over trajectories $\tau = (s_0, a_0, \ldots, s_N)$ be given by $p_\theta(\tau) = p(s_0) \prod_{i=0}^{N-1} \pi_\theta(a_i|s_i) p(s_{i+1}|s_i, a_i)$ and $p_\theta(s_{0:N})$ denote its state-marginal distribution. For simplicity, we assume that the policy is represented as a multivariate normal distribution with a learnable diagonal covariance matrix: $\pi_\theta(a_i|s_i) \sim \mathcal{N}(\mu_{\theta_\mu}(s_i), \Sigma)$, where the mean $\mu_{\theta_\mu}(s_i) = [\mu_{\theta_\mu,1}(s_i), \ldots, \mu_{\theta_\mu,K}(s_i)]^\top$ is modeled by a neural network, $\Sigma = \mathrm{diag}(\sigma_1^2, \ldots, \sigma_K^2)$ is the learnable variance that is independent of states (as in the original TRPO [28] and PPO [30]). Thus, $\theta = [\theta_\mu^\top, \sigma_1, \ldots, \sigma_K]^\top$ is the whole parameters of the policy $\pi_\theta$, and $K = \dim(\mathcal{A})$.

The derivative of the RL objective function $J(\theta) = \mathbb{E}_{\tau \sim p_\theta(\tau)}[R(\tau)]$ can be written as

$$\nabla_\theta J(\theta) = \mathbb{E}_{\tau \sim p_\theta(\tau)}\left[\left(\sum_{i=0}^{N-1} \nabla_\theta \log \pi_\theta(a_i|s_i)\right) R(\tau)\right] \triangleq \mathbb{E}_{\tau \sim p_\theta(\tau)}[G_\theta(\tau)], \quad (3)$$

which is often referred to as the *policy gradient estimator*.

We derive a lower bound for its total variation $\mathrm{tr}[\mathbb{V}_{\tau \sim p_\theta(\tau)}[G_\theta(\tau)]]$, where $\mathbb{V}[X]$ is the variance of a variable $X$ (or the covariance matrix when $X$ is multidimensional), and $\mathrm{tr}$ is the trace operator.

**Theorem 1** *If the environment is stochastic in the sense that for any reparameterized actions $\epsilon_{0:N-1}$, if the variance of returns conditioned on the actions is lower bounded by a small positive constant $c$ (i.e., $\mathbb{V}_{s_{0:N} \sim p_\theta(s_{0:N}|\epsilon_{0:N-1})}[R(\tau)] \geq c > 0$, where $a_i = \mu_{\theta_\mu}(s_i) + \Sigma\epsilon_i$ and $\epsilon_i \overset{i.i.d.}{\sim} \mathcal{N}(0, I)$), it holds that*

$$\mathrm{tr}\left[\mathbb{V}_{\tau \sim p_\theta(\tau)}[G_\theta(\tau)]\right] \geq \frac{Tc}{\delta \cdot \min(\sigma_1^2, \sigma_2^2, \ldots, \sigma_K^2)}. \quad (4)$$

We provide a proof and further discussion in Appendix A. Equation (4) indicates that $\delta \to 0$ can lead the variance of the PG estimator to explode, especially in stochastic environments. We emphasize that the two primary causes of this explosion are the independence of actions and the infinitely growing number of decision steps, both of which correspond to Equation (22) → (23) in Appendix A.

Practically, existing PG algorithms are implemented with standard variance reduction techniques such as reward-to-go policy gradient and baseline functions. However, even if such techniques are applied, the variance of the PG estimator is still likely to explode as $\delta \to 0$ considering the environment's stochasticity, if the learning rate and minibatch size remain the same.

**Challenging exploration.** Furthermore, existing PG methods are prone to perform worse with a low $\delta$ due to the difficulty of exploration. The example in Figure 1 compares 2-D random walks of the same physical time limit with different $\delta$'s. It is easier to get out of the blue box by chance with a high $\delta$ than a low $\delta$. Intuitively, when $\delta \to 0$, the range that the agent can move at each step becomes smaller, which makes it challenging to reach distant states by pure exploration. This originates again from independently sampled actions. We provide a more formal explanation in Appendix B. This exploration problem has also been addressed in Q-learning literature [16, 36], which employs as a remedy autocorrelated noise such as an Ornstein-Uhlenbeck process.

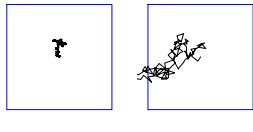

(a) Low $\delta$     (b) High $\delta$

Figure 1: An example of 2-D random walks having the same physical time limit with different $\delta$'s.

**Infinite decision steps.** For a given time limit, the number of decision steps is inversely proportional to $\delta$. This makes the size of the training data required for consuming the same number of episodes increase infinitely as $\delta \to 0$, and thereby impedes training in terms of computational cost.

## 4.2 Durative Actions in Previous Works

For the aforementioned problems with PG methods given a low $\delta$, *durative actions* can be a solution; that is, the policy decides actions only when it is necessary, rather than at every $\delta$. With durative actions, (1) it naturally makes ($\delta$-discretized) actions correlated with one another, and (2) it does not always require infinitely many decision steps even if $\delta \to 0$.

For durative actions, one may modify existing policies to produce both action $a$ and its duration $t$. In discretized continuous-time environments, this approach is practically equivalent to prior methods that produce repetition counts [15, 31], as continuous durations should be converted into the repetitions of time-discretized actions. As one of such action repetition methods, FiGAR-A3C [31] defines the policy as $\pi(a, x|s)$ where $a \in \mathcal{A}$ and $x \in W$. $W$ is a predefined set of action repetition counts; *e.g.*, $W = \{1, 2, \ldots, 30\}$. At every decision step, FiGAR-A3C samples action $a_i$ and repetition count $x_i$ from the policy, and then performs the action $x_i$ times, where $i$ denotes the $i$-th *decision* step. It defines the $n$-step return $\hat{V}^{(n)}(s_i)$ as

$$\hat{V}^{(n)}(s_i) = \sum_{k=0}^{n-1} \gamma^{y_{i+k} - y_i} r_{i+k} + \gamma^{y_{i+n} - y_i} V(s_{i+n}), \tag{5}$$

where the cumulative repetition counts $\{y_i\}$ is defined as $y_0 = 0$ and $y_{i+1} = y_i + x_i$ for $i \geq 0$. The reward $r_i$ at the $i$-th *decision* step is given by the discounted sum of environment rewards over the holding time. While FiGAR-A3C is based on the standard procedure of A3C [20], it is also applicable to other RL algorithms such as TRPO [28] and DDPG [16].

FiGAR can be naturally extended to its continuous variant, which we call FiGAR-C, by replacing $\pi(a, x|s)$ with $\pi(a, t|s)$ where $t \in [0, t_{\max}]$ stands for the *duration* of the action $a$. In $\delta$-discretized environments, it translates the action duration $t$ into $\lceil \frac{t}{\delta} \rceil$ repetition times. FiGAR-C is inherently $\delta$-invariant since it operates on the unit of physical time instead of the discretized time scale.

However, these approaches to action duration have two limitations. First, since it does not consider stopping an action during repetition, it cannot immediately react to unexpected events while repeating an action. This may lead to poor performance in stochastic environments. Second, in contrast to the fact that the optimal policy $\pi(a|s)$ of a fully observable MDP only depends on states $s$, not time $t$ [35], previous methods *only* consider the locality of the time variable $t$, without caring about underlying changes in $s$. This discrepancy can lead to performance degradation, since even during a small time period, $s$ can greatly vary in environments with stochastic or non-continuous dynamics. In the next section, we will demonstrate that such limitations can still bring back the variance explosion problem in these action repetition approaches.

### 4.3 Safe Action Repetition

We propose an alternative approach named *Safe Action Repetition* (SAR) that resolves the limitations of existing action repetition methods. SAR repeats actions based on *state* locality, taking the same action only when states are close.

Following [32], we define a *perturbation set* for a state $s \in \mathcal{S}$ as $\mathbb{B}_\Delta(s, d) = \{s' | \Delta(s, s') \leq d\}$, which corresponds to the closed ball of radius $d$ centered at $s$ using the metric $\Delta$ in the state space. Employing the notion of perturbation sets, we propose the following action repetition scheme. At every decision step, SAR's policy $\pi(a_i, d_i | s_i)$ produces both action $a_i$ and the radius $d_i$ of a perturbation set, which we call the *safe region*. Then, SAR repeats the action $a_i$ only within the safe region $\mathbb{B}_\Delta(s_i, d_i)$, or equivalently

$$\Delta(s, s_i) \leq d_i, \tag{6}$$

where $s$ denotes the current state during repetition. Once the agent goes outside of the safe region, SAR stops action repetition and selects a new action.

Having action durations thresholded by Equation (6) grants two advantages. First, it naturally ensures $\delta$-invariance since action duration is determined by the safe region radius, which is not related to how fine the discretization time scale is. Equation (6) is even completely agnostic to the physical time variable $t$. Second, the agent becomes robust to stochasticity, *e.g.*, encountering an unpredicted event, because such a situation would push the agent's state far away from the safe region, which immediately stops action repetition.

To intuitively differentiate SAR with previous action repetition methods, we illustrate a simple example of a stochastic environment, where previous FiGAR-C fails to maintain an optimal policy due to the variance explosion of the PG estimator, while our method does not. Let us consider the following $\delta$-discretized environment named *AlertThenOff*, whose physical time limit $T$ is 1. The state is $s \in \{0 \text{ (normal)}, 1 \text{ (alerted)}\}$ with $s_0 =$

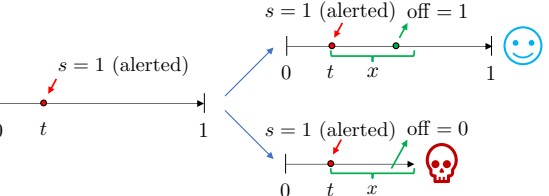

Figure 2: Illustration of *AlertThenOff* environment.

0 (normal). The 2-D action is $a = [\text{off}, \text{num}]^\top$ where $\text{off} \in \{0, 1\}$ and $\text{num} \in \mathbb{R}$. In this environment, $s$ randomly changes once from 0 (normal) to 1 (alerted) at time $t \in [0, 1]$. When $s = 1$ (alerted), we should take an action of $\text{off} = 1$ within $[t, t + x]$ so that $s$ can be back to 0 (normal), where $x > \delta > 0$ is an environment parameter known to us. Otherwise, the environment produces a penalty reward of $-\nu$ (a large negative number) and ends immediately. Conversely, if we set $\text{off} = 1$ when $s = 0$ (normal), it also causes a penalty of $-\nu$ and ends immediately. We illustrate this in Figure 2. The reward (before discretization) is given by $r(t) = f(\text{num})$, where $f$ is an unknown reward function, and its discretized reward is given accordingly to Equation (2). When time reaches $t = 1$, a noisy reward of $\xi \sim \mathcal{N}(0, 1)$ occurs. To sum up, we should perform $\text{off} = 1$ as soon as possible only when we get to know that $s$ becomes 1 (alerted), while we should also perform appropriate num actions that maximize $f(\text{num})$.

To find an optimal policy in this environment, we consider a deterministic policy for off such that $\pi^{\text{off}}(s) = s$, which we already know is optimal, and a stochastic policy for num that $\pi^{\text{num}}(\text{num}|s) \sim \mathcal{N}(\mu, 1)$, where $\mu$ is the policy's parameter. Our goal is thus to find the optimal value of $\mu$. If we assume that the penalty is infinitely large and $f \equiv 0$ for simplicity, we obtain the following result.

**Proposition 2** *In AlertThenOff environment, for the optimal policy $\pi_{\theta_f}$ for FiGAR-C and the optimal policy $\pi_{\theta_s}$ for SAR, the following holds:*

$$\text{tr}\left[\mathbb{V}_{\tau \sim p_{\theta_f}(\tau)}[G_{\theta_f}(\tau)]\right] \to \infty, \quad \text{tr}\left[\mathbb{V}_{\tau \sim p_{\theta_s}(\tau)}[G_{\theta_s}(\tau)]\right] = 2, \tag{7}$$

*when $\delta \to 0$, $x \to 0$, $\nu \to \infty$ and $f \equiv 0$.*

We provide a proof in Appendix C. From Proposition 2, we can conclude that in contrast to FiGAR-C, our SAR policy does not suffer from variance explosion in *AlertThenOff* environment. As the previous approach fails to maintain an optimal policy even in this very simple stochastic environment, it can also be at risk of failure in general stochastic environments. The intuition behind this failure is

that it has no choice but to infinitely shorten action durations in order to optimally handle stochasticity that requires immediate reactions from the agent, which leads both the number of decision steps and the variance of the policy gradient to explode. On the other hand, SAR can be free from variance explosion despite such stochasticity, if SAR sets appropriate safe regions so that such an exigent state locates outside of the safe regions, and thereby the number of decision steps can be bounded.

## 4.4 SAR on Non-Markovian Environments

We derived SAR based on the fact that the optimal policy in a fully observable MDP only depends on states. However, if the Markovian property does not hold (*e.g.*, environments with partially observable MDPs [11] or time limits [23]), the optimal policy might not be fully determined by states alone. In this case, we can additionally incorporate temporal thresholds into SAR by extending the definition of the safe region as follows:

$$\lambda \cdot \Delta(s, s_i) + (1 - \lambda)|t - t_i| \leq d_i, \tag{8}$$

where $t_i$ denotes the time at the $i$-th step, $t$ denotes the current time during repetition, and $0 \leq \lambda \leq 1$ is the coefficient that controls the trade-off between distance and time differences. Note that this variant of SAR, which we call $\lambda$-*SAR*, is $\delta$-invariant too since each term is independent from $\delta$.

# 5 Experiments

We apply SAR to three policy gradient (PG) methods, PPO [30], TRPO [28] and A2C [20], and compare with baseline methods in multiple settings. We first demonstrate the $\delta$-invariance of our method on deterministic continuous control environments, compared to previous $\delta$-invariant algorithms (Section 5.1). We then evaluate SAR on stochastic environments to show its robustness to stochasticity (Section 5.2). We also provide illustrative examples for a better understanding of our method (Section 5.3). We describe the full experimental details in Appendix J.

**Experimental setup.** We test SAR on eight continuous control environments from MuJoCo [37]: InvertedPendulum-v2, InvertedDoublePendulum-v2, Hopper-v2, Walker2d-v2, HalfCheetah-v2, Ant-v2, Reacher-v2 and Swimmer-v2. We mainly compare our method to FiGAR-C described in Section 4.2 because it is the only prior method that is $\delta$-invariant and does not always require infinite decision steps even if $\delta \to 0$, but we also make additional comparisons with other baselines such as DAU [36], ARP [14], modified PPO as well in Section 5.1 and Appendices F.2 and G.

For SAR's distance function in Equation (6), we use $\Delta(s, s_i) = \|\tilde{s} - \tilde{s_i}\|_1 / \dim(\mathcal{S})$, where $\| \cdot \|_1$ is the $\ell_1$ norm and $\tilde{s}$ is the state normalized by its moving average. This distance function corresponds to the average difference in each normalized state dimension, where the normalization permits sharing the hyperparameter $d_{\max}$ for all MuJoCo tasks. We also share $t_{\max}$ in FiGAR-C for all environments. Finally, we impose an upper limit of $t_{\max}$ on the maximum duration of actions in SAR for two reasons: (1) to further stabilize training and (2) to ensure a fair comparison with FiGAR-C by setting the same limit on time duration. We provide an ablation study including an analysis of imposing an upper limit on $t$ in Appendix I.

## 5.1 Results on Deterministic Environments

We first train SAR and FiGAR-C with PPO, TRPO and A2C on the MuJoCo environments, which have deterministic transition dynamics (although they have randomized initial states), with various discretization time scales ranging from $5e - 4$ to $5e - 2$ (details in Appendix J). We use suffixes such as '-PPO' to denote the base PG algorithms. Figure 3 shows the training curves of PPO and TRPO with SAR and with FiGAR-C on the eight MuJoCo environments, where the $x$ and $y$ axes denote the number of decision steps and the total reward, respectively. We provide results on A2C and further comparison between SAR-PPO and PPO in Appendices D and F. Figure 3 shows that while vanilla PG algorithms fail to maintain their performance in lower-$\delta$ settings, SAR is mostly robust to varying $\delta$, often even achieving the best performance with the lowest $\delta$. In a few environments such as Swimmer-v2, SAR's performance becomes slightly worse as $\delta$ decreases, which we hypothesize is because a higher $\delta$ aids the agent in getting out of local optima. Also, SAR exhibits similar or better performance compared to FiGAR-C on most deterministic environments.

For a more comprehensive evaluation, we additionally compare with DAU [36], which is another approach to $\delta$-invariance for Q-learning methods such as DQN [18] and DDPG [16]. Note that

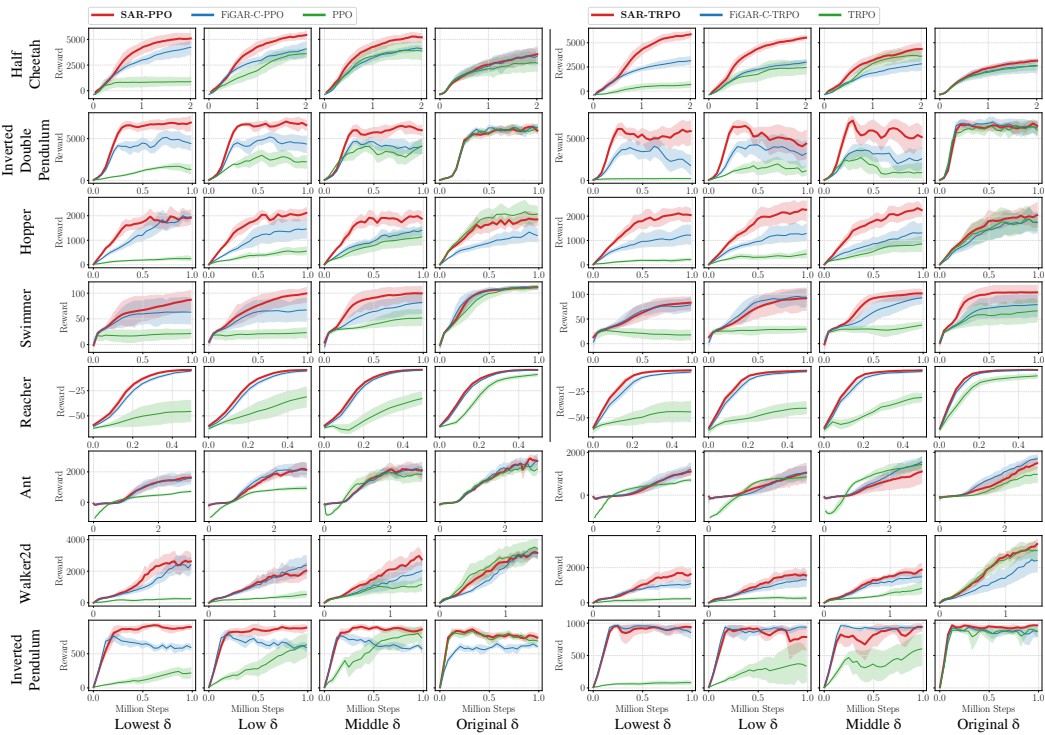

Figure 3: Training curves on deterministic MuJoCo environments with various $\delta$'s (ranging from $5e-4$ to $5e-2$). Shaded areas represent the $95\%$ confidence intervals over eight runs. We compare SAR to FiGAR-C with two base PG algorithms, PPO and TRPO. SAR mostly has $\delta$-invariance, showing similar or even better performance with lower $\delta$'s.

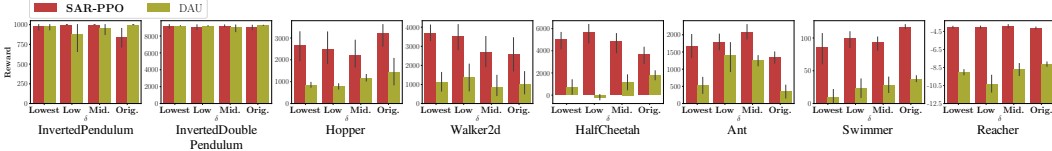

Figure 4: Bar plots comparing the final performance of SAR-PPO to DAU's on deterministic MuJoCo environments with various $\delta$'s. Error bars represent the $95\%$ confidence intervals over eight runs.

SAR-PPO and DAU have different underlying algorithms and training schemes. While DAU is based on DDPG and chooses an action at every environment step, SAR-PPO operates on PPO and makes action decisions only when needed. For the comparison, we use the official implementation of DAU [36]. Figure 4 compares the final performances of both methods with various $\delta$'s at the same physical time (equal to $1e6$ environment steps in the original $\delta$) on each environment. Overall SAR-PPO outperforms DAU and exhibits better $\delta$-invariance in most of the environments. Also, our method requires about $17.6\times$ fewer decision steps on average in the lowest-$\delta$ settings via temporally extended actions.

## 5.2   Results on Stochastic Environments

To demonstrate SAR's robustness with stochastic dynamics, we modify existing MuJoCo environments by adding various types of stochasticity. (1) "External Force": we apply an external force with a standard deviation of $\sigma_{\text{ext}}$ to the agent's body with a probability of $p_{\text{ext}}$ at each decision step. (2) "Strong External Force (Perceptible)": we make external forces perceptible by the agent, which allows it to react to stronger forces with $\sigma_{\text{ext2}} > \sigma_{\text{ext}}$. (3) "Action Noise": we apply noise with a standard deviation of $\sigma_{\text{act}}$ to the action with a probability of $p_{\text{act}}$ at each decision step. Throughout this experiment, we use the lowest-$\delta$ settings. Figure 5 compares SAR-PPO's performance with

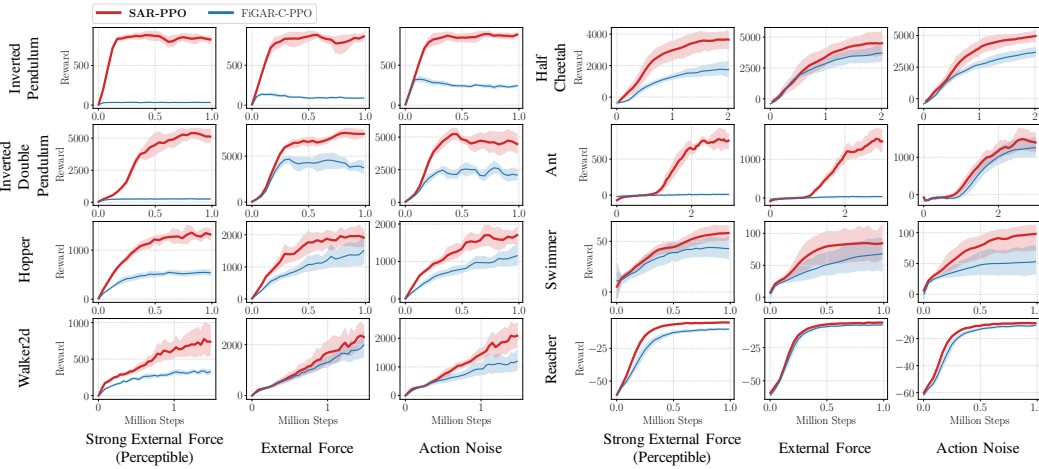

Figure 5: Training curves of SAR-PPO and FiGAR-C-PPO on MuJoCo environments with various types of stochasticity. Shaded areas represent the $95\%$ confidence intervals over eight runs. SAR exhibits strong performance in the presence of stochasticity.

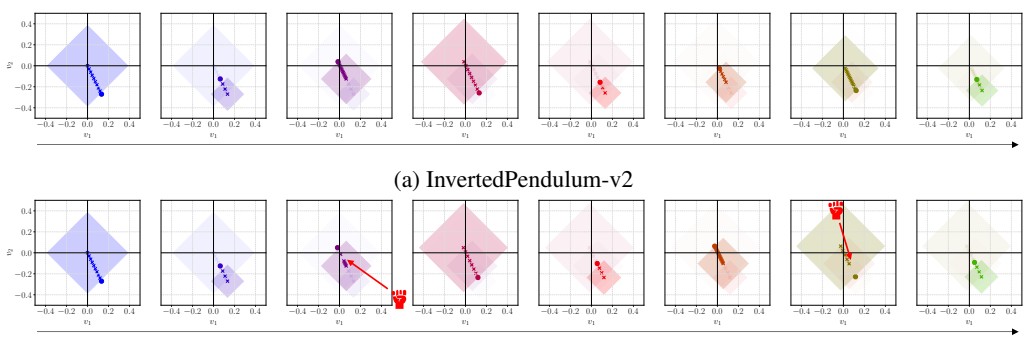

(a) InvertedPendulum-v2

(b) InvertedPendulum-v2 with external forces

Figure 6: Illustration of SAR on InvertedPendulum-v2 with $\delta = 2e - 3$. Our policy produces safe regions (rhombus) within which the agent repeats actions. Circle markers represent the *break points* of action repetitions and red fists represent the external forces applied to the agent.

FiGAR-C-PPO's, and shows that SAR outperforms FiGAR-C on most of the stochastic environments, often exhibiting drastic differences. We provide further details and the comparison on TRPO in Appendices D and J.

## 5.3 Qualitative Analysis of SAR

In order to provide further insights on SAR, we illustrate how SAR works on InvertedPendulum-v2, where the goal is to maintain the balance of a pendulum. The state space consists of four dimensions: two for the position of the agent and the other two for its velocity; *i.e.*, $s = [x_1, x_2, v_1, v_2]$. We demonstrate the behavior of SAR on a 2-D plane. For better interpretability, we slightly modify our method in this experiment: we use only the two (normalized) velocity dimensions for the distance function; that is, the agent stops action repetitions only by its velocity.

Figure 6a illustrates how a trained SAR model performs action repetition based on safe regions, where the $x$ and $y$ axes correspond respectively to $v_1$ and $v_2$, rhombuses represent safe regions, markers (either circle or cross) represent time-discretized states. It can be observed that our policy is learned to produce a large safe region when the agent's speed is low and, conversely, a small safe region when the speed is high, which fits with the intuition because the risk of losing the balance of the pendulum rises as the agent's speed increases.

Additionally, we demonstrate how SAR operates in the presence of stochastic external forces (described in Section 5.2) in Figure 6b. It shows how SAR can quickly and adaptively handle such stochasticity with safe regions by immediately stopping repetitions.

**How SAR learns the adaptive sizes of safe regions?** When the agent's velocity is low, a bigger safe region becomes a low-hanging fruit especially in the early stages of the training where the PG estimator is mostly dominated by immediate rewards (*i.e.*, the accumulated reward within a single action repetition), and thus SAR gravitates toward producing larger safe regions. On the other hand, when the velocity is high and if safe regions are too large, the pendulum would easily lose the balance and thus lead to the end of the episode, which makes SAR in favor of smaller safe regions.

## 6    Conclusion

We proposed Safe Action Repetition (SAR), a novel $\delta$-invariant action repetition method for policy gradient (PG) algorithms. SAR can handle infinitesimal-$\delta$ settings using temporally extended actions without suffering from the variance explosion problem of PG methods, which we proved for general stochastic environments under a certain assumption. It can agilely cope with environment stochasticity via learned safe regions. We exhibited that our method achieves both $\delta$-invariance and robustness to stochasticity.

**Limitations and future directions.** SAR operates in environments where the distance functions can be properly defined in the state spaces. We experimented with the $\ell_1$ norm as SAR's distance function, which might not be applicable to discrete or very high dimensional environments such as vision-based simulations. As such, we expect that combining our method with representation learning techniques [9, 10, 27] would be an interesting future research direction. Also, since SAR in Section 4.3 assumes fully observable MDPs so that it can exploit state locality, there is room for improvement in environments with noisy states or partially observable MDPs. Although we have suggested one possible solution in Section 4.4, this could also be the subject of future research.

## 7    Broader Impact

We expect that our method is especially useful in a variety of real-world situations where the sampling frequencies of simulated environments and real-world physical sensors are different or where the agent could encounter unexpected situations in the environment. However, in spite of the potential positive aspects, practitioners need to pay sufficient attention to various perspectives on their problems and our assumptions when trying to apply our proposed method to real-world problems. For example, in some environments such as autonomous driving, minimizing risk may be more crucial than maximizing rewards, for which they may need to consider incorporating risk-averse methods [26, 33]. Also, they have to examine the degree to which the assumptions made in this work are satisfied in their problems. For instance, the Markovian property may not generally hold in practical settings, in which applying our method as-is might possess potential risks. They should analyze the ramifications of the assumption mismatch and handle them accordingly (see also Section 6). With such considerations, we hope that our research provides new insights toward $\delta$-invariant and robust reinforcement learning.

## Acknowledgements

We thank the anonymous reviewers for their helpful comments. This work was supported by Samsung Advanced Institute of Technology, Brain Research Program by National Research Foundation of Korea (NRF) (2017M3C7A1047860), Institute of Information & communications Technology Planning & Evaluation (IITP) grant funded by the Korea government (MSIT) (No.2019-0-01082, SW StarLab) and Institute of Information & communications Technology Planning & Evaluation (IITP) grant funded by the Korea government (MSIT) (No.2021-0-01343, Artificial Intelligence Graduate School Program (Seoul National University)). Gunhee Kim is the corresponding author.

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
