## A Proof of Theorem 1

In this section, we derive a lower bound for the trace of the covariance of the PG estimator in environments with stochastic dynamics. Recall that $\pi_\theta(a_i|s_i) \sim \mathcal{N}(\mu_{\theta_\mu}(s_i), \Sigma)$ and $p_\theta(\tau) = p(s_0) \prod_{i=0}^{N-1} \pi_\theta(a_i|s_i) p(s_{i+1}|s_i, a_i)$, where the mean $\mu_{\theta_\mu}(s_i) = [\mu_{\theta_\mu,1}(s_i), \ldots, \mu_{\theta_\mu,K}(s_i)]^\top$ (the subscript $k$ in $\mu_{\theta_\mu,k}(s_i)$ denotes the $k$-th element of the vector $\mu_{\theta_\mu}(s_i)$) is modeled by a neural network, $\Sigma = \mathrm{diag}(\sigma_1^2, \ldots, \sigma_K^2)$ is the learnable variance that is independent of states, $\theta = [\theta_\mu^\top, \sigma_1, \ldots, \sigma_K]^\top$ is the whole parameters of the policy $\pi_\theta$, and $K = \dim(\mathcal{A})$.

First, let us consider a single scalar parameter $\vartheta$ that is a bias in the last layer of the mean network $\mu_{\theta_\mu}$, such that $\mu_{\theta_\mu}(s_i) = \mu'_{\theta_{\mu'}}(s_i) + [b_1, b_2, \ldots, b_K]^\top$ and w.l.o.g. $\sigma_1 = \min(\sigma_1, \sigma_2, \ldots, \sigma_K)$ and $\vartheta \triangleq b_1$, where $[b_1, \ldots, b_K]$ is the set of bias parameters in the last layer and $\mu'_{\theta_{\mu'}}(s_i)$ denotes the remainder of the mean network. Then, the following holds:

$$\mathrm{tr}\left[ \mathbb{V}_{\tau \sim p_\theta(\tau)} \left[ \left( \sum_{i=0}^{N-1} \nabla_\theta \log \pi_\theta(a_i|s_i) \right) R(\tau) \right] \right] \tag{9}$$

$$\geq \mathbb{V}_{\tau \sim p_\theta(\tau)} \left[ \left( \sum_{i=0}^{N-1} \frac{\partial}{\partial \vartheta} \log \pi_\theta(a_i|s_i) \right) R(\tau) \right] \tag{10}$$

$$= \mathbb{V}_{\tau \sim p_\theta(\tau)} \left[ \left( \sum_{i=0}^{N-1} \frac{\partial}{\partial \vartheta} \sum_{k=1}^K \left( -\frac{1}{2\sigma_k^2}(a_{i,k} - \mu_{\theta_\mu,k}(s_i))^2 - \frac{1}{2} \log(2\pi\sigma_k^2) \right) \right) R(\tau) \right] \tag{11}$$

$$= \mathbb{V}_{\tau \sim p_\theta(\tau)} \left[ \left( \sum_{i=0}^{N-1} \sum_{k=1}^K \frac{1}{\sigma_k^2}(a_{i,k} - \mu_{\theta_\mu,k}(s_i)) \frac{\partial}{\partial \vartheta} \mu_{\theta_\mu,k}(s_i) \right) R(\tau) \right] \tag{12}$$

$$= \mathbb{V}_{\tau \sim p_\theta(\tau)} \left[ \left( \sum_{i=0}^{N-1} \frac{1}{\sigma_1^2}(a_{i,1} - \mu_{\theta_\mu,1}(s_i)) \right) R(\tau) \right]. \tag{13}$$

By reparameterizing the actions as $a_{i,k} = \mu_{\theta_\mu,k}(s_i) + \sigma_k \epsilon_{i,k}$ for all $0 \leq i \leq N-1$ and $1 \leq k \leq K$, where $\{\epsilon_{i,k}\} \overset{\text{i.i.d.}}{\sim} \mathcal{N}(0,1)$ (we use the simplified notation $\{\epsilon_{i,k}\}$ to denote $(\epsilon_{0,1}, \ldots, \epsilon_{0,K}, \epsilon_{1,1}, \ldots, \epsilon_{N-1,K})$), we obtain

$$\mathbb{V}_{\tau \sim p_\theta(\tau)} \left[ \left( \sum_{i=0}^{N-1} \frac{1}{\sigma_1^2}(a_{i,1} - \mu_{\theta_\mu,1}(s_i)) \right) R(\tau) \right] \tag{14}$$

$$= \mathbb{V}_{\substack{\{\epsilon_{i,k}\} \overset{\text{i.i.d.}}{\sim} \mathcal{N}(0,1) \\ s_{0:N} \sim p_\theta(s_{0:N}|\{\epsilon_{i,k}\})}} \left[ \left( \sum_{i=0}^{N-1} \frac{\epsilon_{i,1}}{\sigma_1} \right) R(\tau) \right]. \tag{15}$$

We then use the law of total variance to decompose Equation (15) as follows:

$$\mathop{\mathbb{V}}_{\substack{\{\epsilon_{i,k}\} \overset{\text{i.i.d.}}{\sim} \mathcal{N}(0,1) \\ s_{0:N} \sim p_\theta(s_{0:N}|\{\epsilon_{i,k}\})}} \left[ \left( \sum_{i=0}^{N-1} \frac{\epsilon_{i,1}}{\sigma_1} \right) R(\tau) \right] \tag{16}$$

$$= \mathop{\mathbb{V}}_{\{\epsilon_{i,k}\} \overset{\text{i.i.d.}}{\sim} \mathcal{N}(0,1)} \left[ \mathop{\mathbb{E}}_{s_{0:N} \sim p_\theta(s_{0:N}|\{\epsilon_{i,k}\})} \left[ \left( \sum_{i=0}^{N-1} \frac{\epsilon_{i,1}}{\sigma_1} \right) R(\tau) \right] \right]$$

$$+ \mathop{\mathbb{E}}_{\{\epsilon_{i,k}\} \overset{\text{i.i.d.}}{\sim} \mathcal{N}(0,1)} \left[ \mathop{\mathbb{V}}_{s_{0:N} \sim p_\theta(s_{0:N}|\{\epsilon_{i,k}\})} \left[ \left( \sum_{i=0}^{N-1} \frac{\epsilon_{i,1}}{\sigma_1} \right) R(\tau) \right] \right] \tag{17}$$

$$\geq \mathop{\mathbb{E}}_{\{\epsilon_{i,k}\} \overset{\text{i.i.d.}}{\sim} \mathcal{N}(0,1)} \left[ \mathop{\mathbb{V}}_{s_{0:N} \sim p_\theta(s_{0:N}|\{\epsilon_{i,k}\})} \left[ \left( \sum_{i=0}^{N-1} \frac{\epsilon_{i,1}}{\sigma_1} \right) R(\tau) \right] \right] \tag{18}$$

$$= \mathop{\mathbb{E}}_{\{\epsilon_{i,k}\} \overset{\text{i.i.d.}}{\sim} \mathcal{N}(0,1)} \left[ \frac{(\sum_{i=0}^{N-1} \epsilon_{i,1})^2}{\sigma_1^2} \mathop{\mathbb{V}}_{s_{0:N} \sim p_\theta(s_{0:N}|\{\epsilon_{i,k}\})} [R(\tau)] \right]. \tag{19}$$

Now we apply the following assumption:

$$\forall \{\epsilon_{i,k}\} \quad \mathop{\mathbb{V}}_{s_{0:N} \sim p_\theta(s_{0:N}|\{\epsilon_{i,k}\})} [R(\tau)] \geq c, \tag{20}$$

where $c$ is a small constant greater than 0. This assumption states that the environment is inherently stochastic in the sense that its return has a variance of at least $c$ even if conditioned on the reparameterized actions $\{\epsilon_{i,k}\}$ (or equivalently, $\mathbb{V}[R(\tau)]$ is greater than or equal to $c$ even if the *random seed* used for sampling actions is fixed). Note that environments with deterministic transition dynamics, such as the MuJoCo environments, could also satisfy this assumption, considering the stochasticity of the initial state: $s_0 \sim p(s_0)$ (although a perfect baseline could cancel out the initial stochasticity in such environments).

Using this assumption, Equation (19) can be rewritten as

$$\mathop{\mathbb{E}}_{\{\epsilon_{i,k}\} \overset{\text{i.i.d.}}{\sim} \mathcal{N}(0,1)} \left[ \frac{(\sum_{i=0}^{N-1} \epsilon_{i,1})^2}{\sigma_1^2} \mathop{\mathbb{V}}_{s_{0:N} \sim p_\theta(s_{0:N}|\{\epsilon_{i,k}\})} [R(\tau)] \right] \tag{21}$$

$$\geq \mathop{\mathbb{E}}_{\{\epsilon_{i,k}\} \overset{\text{i.i.d.}}{\sim} \mathcal{N}(0,1)} \left[ \frac{(\sum_{i=0}^{N-1} \epsilon_{i,1})^2}{\sigma_1^2} c \right] \tag{22}$$

$$= \frac{Nc}{\sigma_1^2} \tag{23}$$

$$= \frac{Tc}{\delta \sigma_1^2} \tag{24}$$

$$= \frac{Tc}{\delta \cdot \min(\sigma_1^2, \sigma_2^2, \ldots, \sigma_K^2)}. \tag{25}$$

From Equation (25), we can conclude that $\delta \to 0$ leads the variance of the PG estimator to explode in stochastic environments.

As a side note, if we leave the other term when decomposing Equation (15), we obtain

$$\underset{\substack{\{\epsilon_{i,k}\} \overset{\text{i.i.d.}}{\sim} \mathcal{N}(0,1) \\ s_{0:N} \sim p_\theta(s_{0:N}|\{\epsilon_{i,k}\})}}{\mathbb{V}} \left[ \left( \sum_{i=0}^{N-1} \frac{\epsilon_{i,1}}{\sigma_1} \right) R(\tau) \right] \tag{26}$$

$$= \underset{\{\epsilon_{i,k}\} \overset{\text{i.i.d.}}{\sim} \mathcal{N}(0,1)}{\mathbb{V}} \left[ \mathbb{E}_{s_{0:N} \sim p_\theta(s_{0:N}|\{\epsilon_{i,k}\})} \left[ \left( \sum_{i=0}^{N-1} \frac{\epsilon_{i,1}}{\sigma_1} \right) R(\tau) \right] \right]$$

$$+ \underset{\{\epsilon_{i,k}\} \overset{\text{i.i.d.}}{\sim} \mathcal{N}(0,1)}{\mathbb{E}} \left[ \underset{s_{0:N} \sim p_\theta(s_{0:N}|\{\epsilon_{i,k}\})}{\mathbb{V}} \left[ \left( \sum_{i=0}^{N-1} \frac{\epsilon_{i,1}}{\sigma_1} \right) R(\tau) \right] \right] \tag{27}$$

$$\geq \underset{\{\epsilon_{i,k}\} \overset{\text{i.i.d.}}{\sim} \mathcal{N}(0,1)}{\mathbb{V}} \left[ \mathbb{E}_{s_{0:N} \sim p_\theta(s_{0:N}|\{\epsilon_{i,k}\})} \left[ \left( \sum_{i=0}^{N-1} \frac{\epsilon_{i,1}}{\sigma_1} \right) R(\tau) \right] \right] \tag{28}$$

$$= \underset{\{\epsilon_{i,k}\} \overset{\text{i.i.d.}}{\sim} \mathcal{N}(0,1)}{\mathbb{V}} \left[ \left( \sum_{i=0}^{N-1} \frac{\epsilon_{i,1}}{\sigma_1} \right) \mathbb{E}_{s_{0:N} \sim p_\theta(s_{0:N}|\{\epsilon_{i,k}\})} \left[ R(\tau) \right] \right]. \tag{29}$$

From this, we can speculate that even in completely deterministic environments, $\delta \to 0$ is also likely to cause variance explosion (*e.g.*, consider a simple setting with $R(\tau) = 1$), but generalizing this may require more sophisticated assumptions, which we leave for future work.

## B    Concrete Example of the Challenging Exploration Problem

In this section, we illustrate the difficulty of exploration with a low $\delta$. Let us consider the following simple continuous-time MDP defined as

$$T = 2 \tag{30}$$
$$\gamma = 1 \tag{31}$$
$$s(t) \in \mathbb{R}^2 \tag{32}$$
$$a(t) \in \{-1, +1\} \tag{33}$$
$$s(0) = [0, 0]^\top \tag{34}$$
$$F(s(t), a(t)) = [a(t), 1]^\top, \tag{35}$$

where $T$ denotes the physical time limit.

Its discretized MDP with a discretization time scale $\delta = \frac{2}{N}$, which equally divides the total duration by $N$, is defined as follows:

$$\tau = (s_0, a_0, \ldots, s_N) \tag{36}$$
$$s_0 = [0, 0]^\top \tag{37}$$
$$s_{i+1} = s_i + \left[ \frac{2}{N} a_i, \frac{2}{N} \right]^\top \tag{38}$$
$$r(s_i, a_i) = \mathbb{1}_{\{|s_{i,0}| \geq 1 \text{ and } s_{i,1} \geq 2\}}, \tag{39}$$

where $\mathbb{1}$ denotes the indicator function and we additionally define the reward function $r(s_i, a_i)$.

Let us assume that the initial policy $\pi(a_i|s_i)$ follows the uniform distribution such that $\pi(a_i = -1|s_i) = \pi(a_i = +1|s_i) = \frac{1}{2}$ for all $i$. Intuitively, this corresponds to a simple 1-D random walk process, where the first dimension of the state denotes the agent's position, the second dimension denotes the current time, and a positive reward occurs if the final position $s_{N,0}$ of the agent is located outside of the interval $(-1, 1)$.

When $\delta \to 0$, we can compute the probability that the agent gets a positive reward with the policy $\pi$ as follows:

$$P(R_\pi(\tau) > 0) = 1 - P\left(\frac{1}{4}N < X < \frac{3}{4}N\right) \qquad \text{where } X \sim B\left(N, \frac{1}{2}\right) \tag{40}$$

$$\approx 1 - P\left(\frac{1}{4}N < Y < \frac{3}{4}N\right) \qquad \text{where } Y \sim \mathcal{N}\left(\frac{N}{2}, \frac{N}{4}\right) \tag{41}$$

$$= 1 - P\left(-\frac{\sqrt{N}}{2} < Z < \frac{\sqrt{N}}{2}\right) \qquad \text{where } Z \sim \mathcal{N}(0, 1) \tag{42}$$

$$\to 0 \qquad \text{as } N = \frac{2}{\delta} \to \infty, \tag{43}$$

where $R_\pi(\tau)$ denotes the random variable corresponding to the return of the trajectory obtained by the policy $\pi$, $B$ denotes the binomial distribution and $\mathcal{N}$ denotes the normal distribution. We use the normal approximation of the binomial distribution because $N$ becomes sufficiently large as $\delta \to 0$.

Equation (43) shows that when the discretization time scale is infinitesimal, it is impossible for the initial random policy to discover a state that produces a positive reward.

## C   Proof of Proposition 2

Recall that we consider the setting with $\delta \to 0$, $x \to 0$, $\delta < x$, $\nu \to \infty$ and $f(\mathrm{num}) = 0$ in the *AlertThenOff* environment; thus the return $R(\tau)$ is given by $\xi$. We first discuss the optimal policy $\pi_{\theta_f}$ for FiGAR-C. Its optimal policy for $t$, $\pi_{\theta_f}^t(t|s)$, should produce $t \leq x$ because otherwise it has the risk of ending up with $-\nu$ reward, which is not an optimum. Therefore, we assume that its (stochastic) duration policy $\pi_{\theta_f}^t(t|s)$, which is parameterized by $\mu_t$, always produces durations that are less than or equal to $x$. The whole parameters of FiGAR-C's policy become $\theta_f = [\mu, \mu_t^\top]$, and $\pi_{\theta_f}$ (which consists of $\pi_{\theta_f}^{\mathrm{off}}$, $\pi_{\theta_f}^{\mathrm{num}}$ and $\pi_{\theta_f}^t$) produces deterministic actions for off and stochastic actions for num and $t$. Now we compute a lower bound for the variance of the PG estimator:

$$\mathrm{tr}\left[\mathbb{V}_{\tau \sim p_{\theta_f}(\tau)}\left[G_{\theta_f}(\tau)\right]\right] \tag{44}$$

$$= \mathrm{tr}\left[\mathbb{V}_{\tau \sim p_{\theta_f}(\tau)}\left[\left(\sum_{i=0}^{N-1} \nabla_{\theta_f} \log \pi_{\theta_f}(\mathrm{num}_i, t_i|s_i)\right) R(\tau)\right]\right] \tag{45}$$

$$\geq \mathbb{V}_{\tau \sim p_{\theta_f}(\tau)}\left[\left(\sum_{i=0}^{N-1} \frac{\partial}{\partial \mu} \log \pi_{\theta_f}^{\mathrm{num}}(\mathrm{num}_i|s_i)\right) R(\tau)\right] \tag{46}$$

$$= \mathbb{V}_{\substack{\{\epsilon_i\} \overset{\mathrm{i.i.d.}}{\sim} \mathcal{N}(0,1) \\ s_{0:N} \sim p_{\theta_f}(s_{0:N}|\{\epsilon_i\})}}\left[\left(\sum_{i=0}^{N-1} \epsilon_i\right) R(\tau)\right] \tag{47}$$

$$\geq \mathbb{E}_{\{\epsilon_i\} \overset{\mathrm{i.i.d.}}{\sim} \mathcal{N}(0,1)}\left[\left(\sum_{i=0}^{N-1} \epsilon_i\right)^2 \mathbb{V}_{s_{0:N} \sim p_{\theta_f}(s_{0:N}|\{\epsilon_i\})}[R(\tau)]\right] \tag{48}$$

$$= \mathbb{E}_{\{\epsilon_i\} \overset{\mathrm{i.i.d.}}{\sim} \mathcal{N}(0,1)}\left[\left(\sum_{i=0}^{N-1} \epsilon_i\right)^2 \mathbb{V}_{\xi \sim \mathcal{N}(0,1)}[\xi]\right] \tag{49}$$

$$= N \geq \frac{1}{x}, \tag{50}$$

where $N$ denotes the number of decision steps. We reparameterize actions as in Equation (15) and use the law of total variance. For simplicity, we assume that FiGAR-C's duration policy is stochastic, but Equation (50) still holds even if the duration policy is deterministic or fixed (in this case, $\theta_f$ becomes $[\mu]$) due to the inequality in Equation (46).

From Equation (50), we can find that when $x \to 0$, both the variance of the policy gradient and the number of decision steps explode to infinity.

On the other hand, let us consider SAR's optimal policy $\pi_{\theta_s}$. If we set $\Delta(s_1, s_2) = |s_1 - s_2|$, one of the optimal (deterministic) policies for $d$ can simply be $\pi_{\theta_s}^d(s) = \frac{1}{2}$. In this case, the whole parameters of SAR's policy become $\theta_s = [\mu]$, and $\pi_{\theta_s}$ (which consists of $\pi_{\theta_s}^{\text{off}}$, $\pi_{\theta_s}^{\text{num}}$ and $\pi_{\theta_s}^d$) produces stochastic actions for num and deterministic actions for off and $d$. Also, $N$ becomes 2, as it stops an action only once when $s$ changes to 1 (alerted). We can then compute the variance of the PG estimator as follows:

$$\text{tr}\left[\mathbb{V}_{\tau \sim p_{\theta_s}(\tau)}\left[G_{\theta_s}(\tau)\right]\right] \tag{51}$$

$$= \mathbb{V}_{\tau \sim p_{\theta_s}(\tau)}\left[\left(\sum_{i=0}^{N-1} \nabla_{\theta_s} \log \pi_{\theta_s}^{\text{num}}(\text{num}_i | s_i)\right) R(\tau)\right] \tag{52}$$

$$= \mathop{\mathbb{V}}_{\substack{\{\epsilon_i\} \overset{\text{i.i.d.}}{\sim} \mathcal{N}(0,1) \\ s_{0:N} \sim p_{\theta_s}(s_{0:N} | \{\epsilon_i\})}}\left[\left(\sum_{i=0}^{N-1} \epsilon_i\right) R(\tau)\right] \tag{53}$$

$$= \mathbb{V}_{\{\epsilon_i\} \overset{\text{i.i.d.}}{\sim} \mathcal{N}(0,1), \xi \sim \mathcal{N}(0,1)}\left[\left(\sum_{i=0}^{N-1} \epsilon_i\right) \xi\right] \tag{54}$$

$$= \mathbb{V}_{\{\epsilon_i\} \overset{\text{i.i.d.}}{\sim} \mathcal{N}(0,1), \xi \sim \mathcal{N}(0,1)}\left[(\epsilon_0 + \epsilon_1) \xi\right] \tag{55}$$

$$= 2. \tag{56}$$

Therefore, we conclude that the optimal policy for SAR does not suffer from either variance explosion or infinite decision steps in *AlertThenOff* environment, even if $x \to 0$.

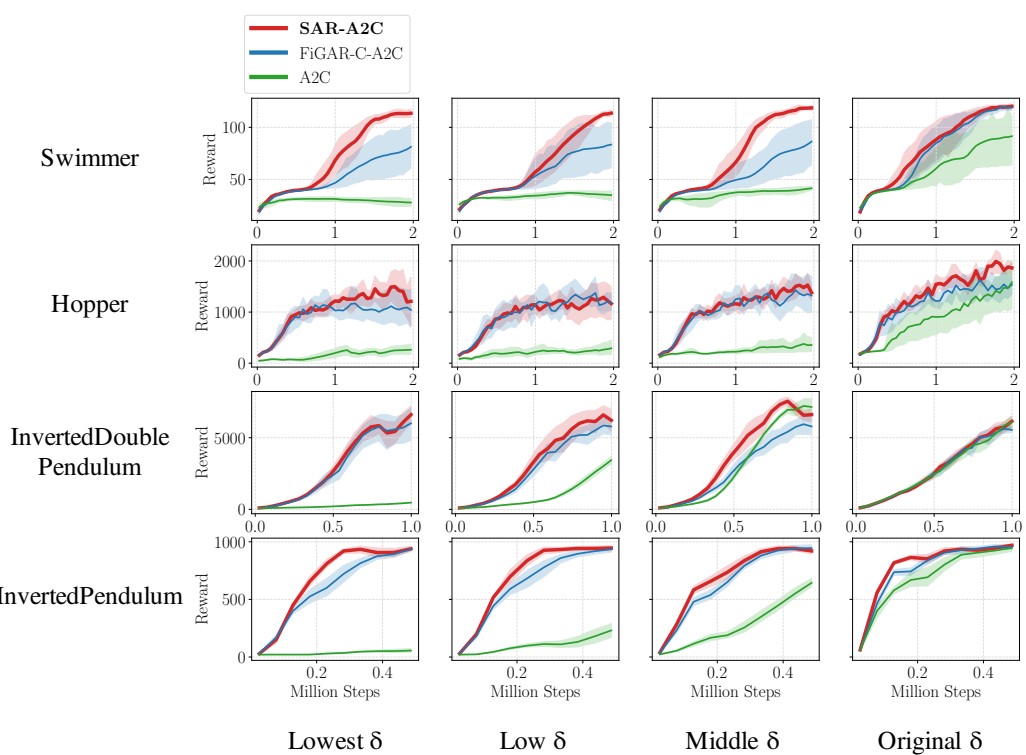

Figure 7: Training curves of SAR-A2C, FiGAR-C-A2C and A2C on four deterministic MuJoCo environments with various $\delta$'s. Shaded areas represent the 95% confidence intervals over eight runs.

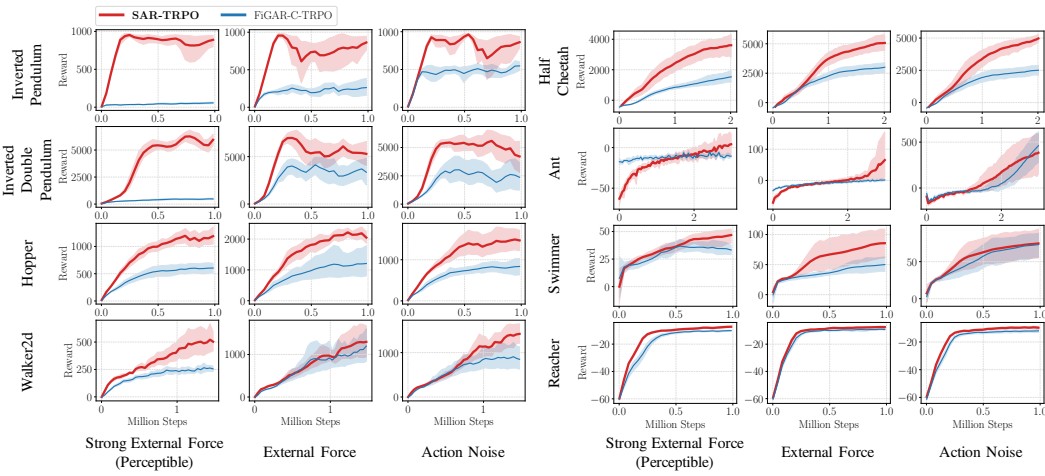

Figure 8: Training curves of SAR-TRPO and FiGAR-C-TRPO on eight MuJoCo environments with various types of stochasticity. Shaded areas represent the 95% confidence intervals over eight runs.

# D    Additional Results

**Deterministic Environments.** We train SAR-A2C, FiGAR-C-A2C, A2C on the four environments of Swimmer-v2, Hopper-v2, InvertedDoublePendulum-v2 and InvertedPendulum-v2, whose results are shown in Figure 7. We find that A2C struggles to perform well on complex environments such as Ant-v2. SAR mostly shows $\delta$-invariance, outperforming the baselines in most of the environments.

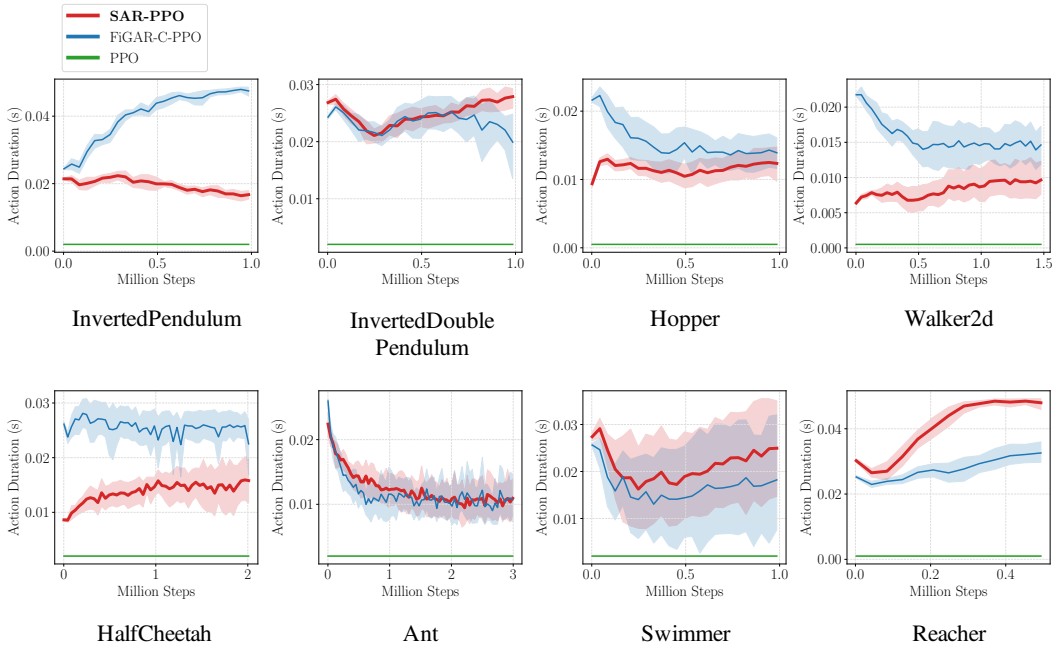

Figure 9: Changes in the average action durations of SAR-PPO, FiGAR-C-PPO and PPO on eight deterministic MuJoCo environments with the lowest-$\delta$ settings. Shaded areas represent the $95\%$ confidence intervals over eight runs.

**Stochastic Environments.** We also provide the result comparing SAR-TRPO to FiGAR-C-TRPO on eight stochastic MuJoCo environments in Figure 8. As in the case of the PPO baseline, SAR-TRPO mostly demonstrates stronger performance than FiGAR-C-TRPO.

**Average action duration.** We provide how the average action durations of SAR-PPO, FiGAR-C-PPO and PPO change as they are trained. Figure 9 demonstrates the results on eight MuJoCo environments with the lowest-$\delta$ settings.

# E   Experiments with Varying Stochasticity Levels

To further examine how SAR and FiGAR-C evolves as the stochasticity level increases, we perform an experiment on stochastic InvertedPendulum-v2 ($\delta = 0.002$) with external forces. We train SAR-PPO and FiGAR-C-PPO on the environment with $p_{\text{ext}} \in \{0.025, 0.05, 0.1, 0.2\}$, where $p_{\text{ext}}$ denotes the probability of an external force being applied. Figure 10 shows the plots of the average reward and the learned (normalized) action duration or safe region radius of each setting. From the second row, we can observe that the learned action duration decreases as the stochasticity level increases in FiGAR-C, while such shrinkage does not happen in SAR. This is because FiGAR-C should reduce action durations when the stochasticity level increases in order to quickly respond to unexpected events. Since FiGAR-C is unaware of underlying state changes, its best strategy is to shorten the duration of actions to be more responsive. On the other hand, SAR does not necessarily shrink the size of safe regions even if the stochasticity level increases because it can easily detect the presence of unexpected events by appropriately setting its safe region sizes. As a result, SAR can handle stochasticity more robustly as well as preventing the variance explosion problem caused by too short action durations.

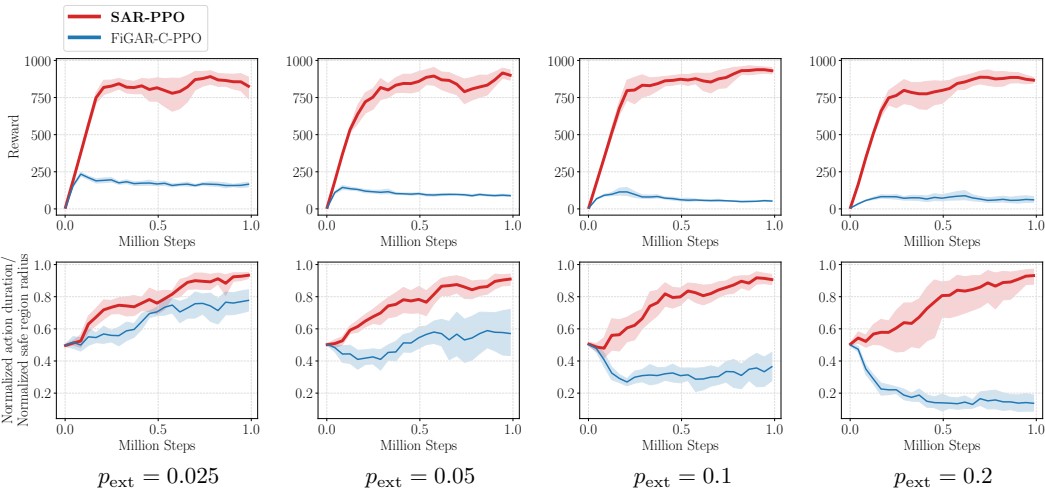

Figure 10: Training curves of SAR-PPO and FiGAR-PPO on InvertedPendulum-v2 with the lowest-$\delta$ setting, in which the stochasticity level $p_{\text{ext}}$ varies from $0.025$ to $0.2$. The first row shows the average performance and the second row shows the average normalized action duration (FiGAR-C-PPO) or safe region radius (SAR-PPO). Shaded areas represent the $95\%$ confidence intervals over eight runs.

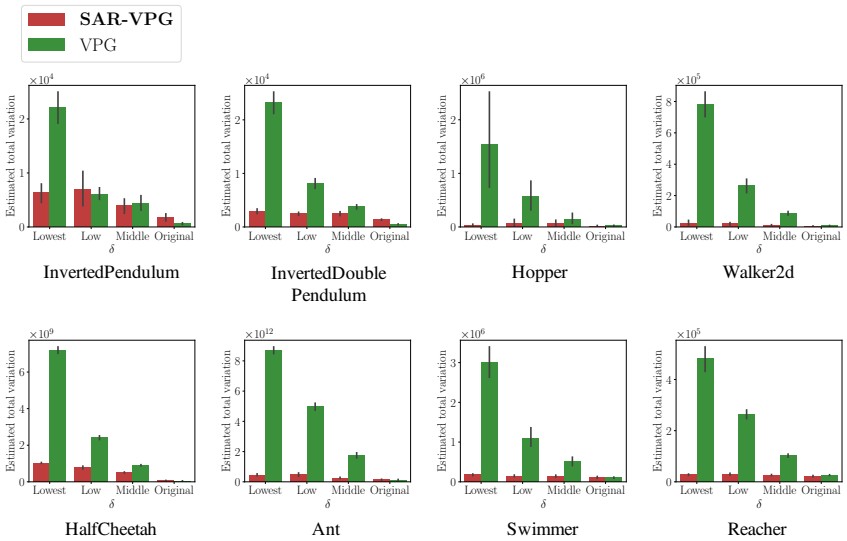

Figure 11: Bar plots showing the estimated total variations $\mathrm{tr}[\hat{\mathbb{V}}_{\tau \sim p_\theta(\tau)}[G_\theta(\tau)]]$ of SAR-VPG and VPG on eight deterministic MuJoCo environments with various $\delta$'s. We estimate the total variation with the initial policy. Error bars represent the 95% confidence intervals over eight runs.

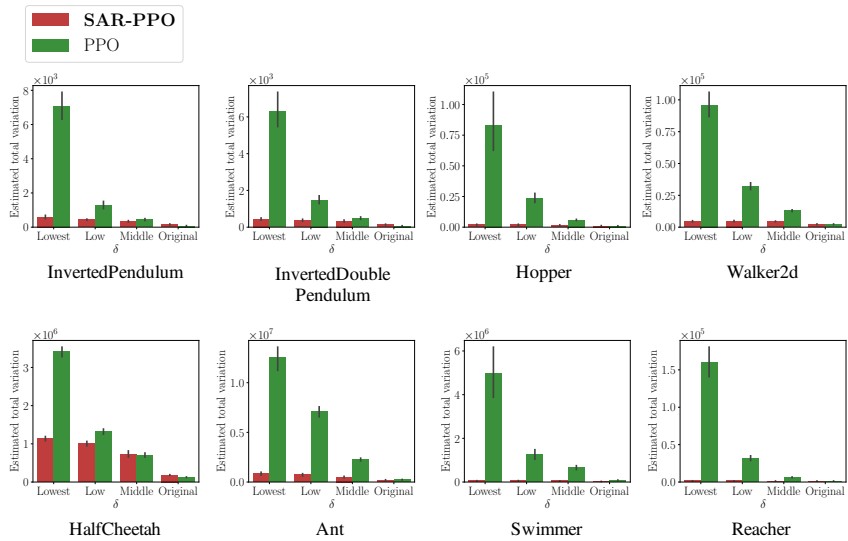

Figure 12: Bar plots showing the estimated total variations $\mathrm{tr}[\hat{\mathbb{V}}_{\tau \sim p_\theta(\tau)}[G_\theta(\tau)]]$ of SAR-PPO and PPO on eight deterministic MuJoCo environments with various $\delta$'s. We estimate the total variation with the initial policy. Error bars represent the 95% confidence intervals over eight runs.

## F Further Demonstrations of PG Methods' Failure with a Low $\delta$

### F.1 Variance Explosion of the PG Estimator

As shown in Theorem 1, policy gradient methods are subjected to the variance explosion problem with an exceedingly small $\delta$. In this section, we empirically demonstrate this phenomenon on eight MuJoCo environments. We estimate the total variation $\mathrm{tr}[\hat{\mathbb{V}}_{\tau \sim p_\theta(\tau)}[G_\theta(\tau)]]$ with the two baseline policy gradient methods: Vanilla Policy Gradient (VPG) and PPO. In VPG, we do not use any technique for variance reduction such as value functions and reward-to-go policy gradient; hence, the formula for its gradient estimator is identical to Equation (3). In PPO, we employ the same implementation used for our main results, including multiple variance reduction techniques such as

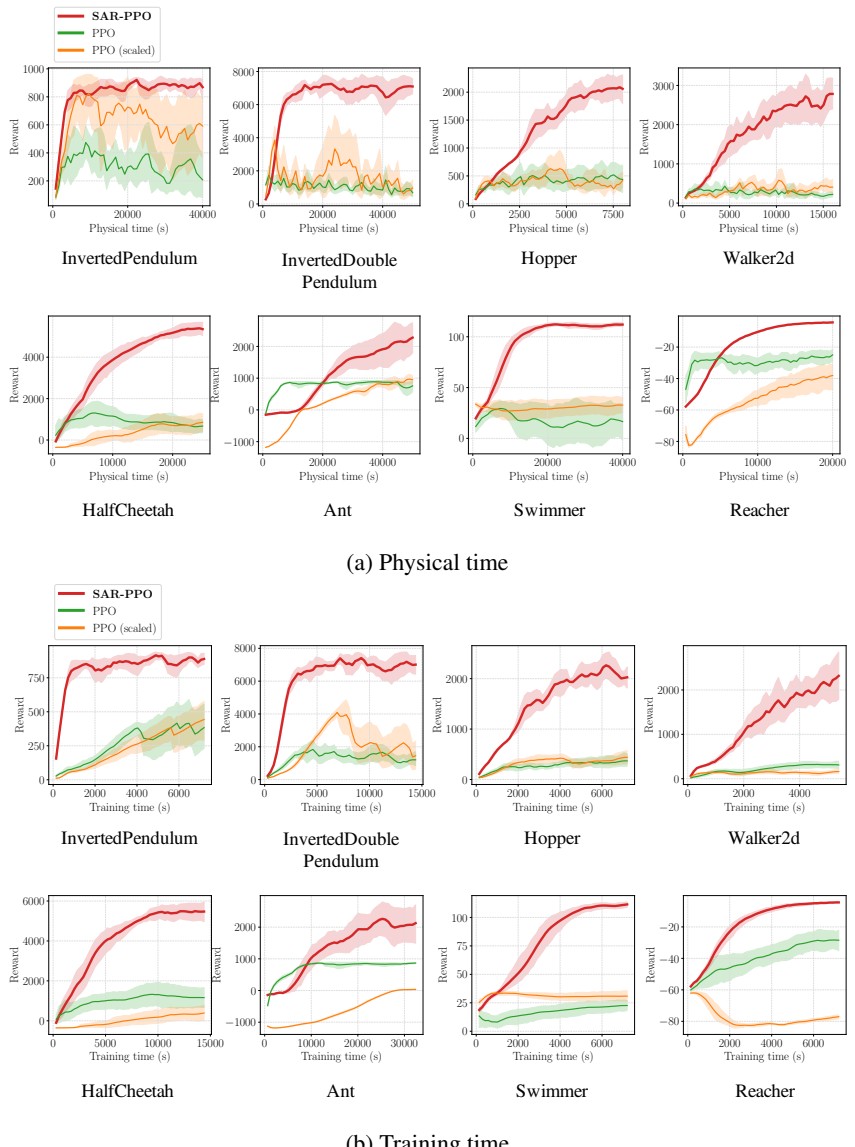

Figure 13: Training curves of SAR-PPO and PPO's variants with respect to (a) physical time and (b) training time for $x$-axis on eight deterministic MuJoCo environments with the lowest-$\delta$ settings. Shaded areas represent the $95\%$ confidence intervals over eight runs.

the GAE [29]. We estimate the total variation with 100 randomly sampled trajectories (after sampling 10 trajectories for an initial burn-in phase) on each of eight randomly initialized policies. Figures 11 and 12 show the estimated total variations of both the baseline methods and SAR with various $\delta$'s. These results confirm that variance explosion empirically occurs on both VPG and PPO with lower-$\delta$ settings, whereas our SAR method can alleviate such a problem.

## F.2 Further Demonstrations of PPO with a Low $\delta$

We verify both theoretically (Section 4.1) and empirically (Appendix F.1) that PG methods suffer from variance explosion if the learning rate and minibatch size remain the same. In this section, we show that PG methods still fail in low-$\delta$ settings even if such parameters are properly scaled, possibly due to the difficulty of exploration (Section 4.1). We additionally test a variant of PPO ("PPO (scaled)") with a learning rate scaled by $\delta/\delta_0$ as in Wawrzynski [38], where $\delta_0$ denotes the original discretization time scale of each environment. Also, in order to compare them on multiple

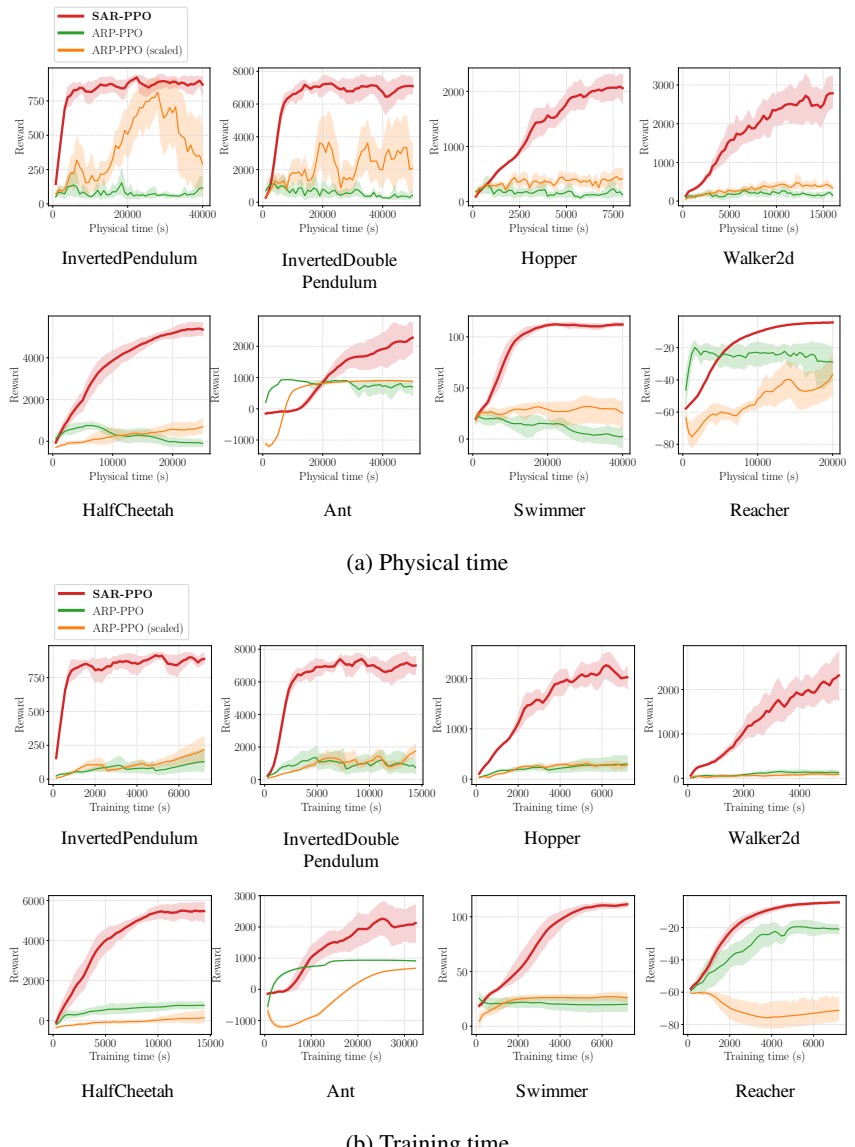

Figure 14: Training curves of SAR-PPO and ARP-PPO's variants with respect to (a) physical time and (b) training time for $x$-axis on eight deterministic MuJoCo environments with the lowest-$\delta$ settings. Shaded areas represent the $95\%$ confidence intervals over eight runs.

criteria, we plot the results on the two $x$-axes of *physical* time and *training* time, where physical time indicates the time elapsed in the simulated environment and training time indicates the time elapsed in the real world for training. Figure 13 demonstrates the training curves on deterministic MuJoCo environments with the lowest-$\delta$ settings. We observe that both PPO and the scaled PPO variant struggle with small discretization time scales. On the contrary, SAR-PPO exhibits strong performance compared to the baseline PPO methods. Furthermore, as revealed by the comparison between the physical time curve and the training time curve of InvertedPendulum-v2, the result suggests that our method significantly facilitates training via action repetition.

## G   Comparison with Autoregressive Policies

We make an additional comparison with autoregressive policies (ARPs) [14], which use autoregressive processes that could prevent the variance explosion problem with a low $\delta$. An ARP uses the

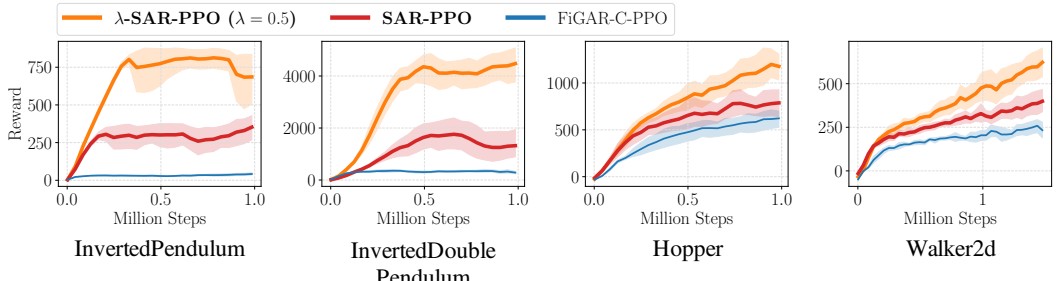

Figure 15: Training curves of $\lambda$-SAR-PPO ($\lambda = 0.5$), SAR-PPO and FiGAR-C-PPO on four stochastic POMDP environments with the lowest-$\delta$ settings. Shaded areas represent the 95% confidence intervals over eight runs. $\lambda$-SAR shows better performance compared to the others in these POMDP settings.

autoregressive noise process to sample actions so that the actions can be temporally correlated. It has two main hyperparameters: $p_{\text{ord}}$ and $\alpha$, where $p_{\text{ord}}$ is the order of the autoregressive process and $\alpha$ controls its temporal smoothness ($\alpha = 0$ corresponds to the white Gaussian noise).

We compare SAR-PPO to ARPs trained with PPO ("ARP-PPO") as well as its variant ("ARP-PPO (scaled)") with the scaled learning rate specified in Appendix F.2. For ARP-PPO and ARP-PPO (scaled), we respectively perform hyperparameter search over $p \in \{1, 3\}$ and $\alpha \in \{0.3, 0.5, 0.8, 0.95\}$. We individually tune the hyperparameters on each environment, while we share the hyperparameter $d_{\max} = 0.5$ across all the environments in the case of SAR. The hyperparameters used for ARPs are as follows:

- Ant-v2: $p = 3$, $\alpha = 0.5$ for ARP-PPO and $p = 3$, $\alpha = 0.5$ for ARP-PPO (scaled).
- HalfCheetah-v2: $p = 3$, $\alpha = 0.3$ for ARP-PPO and $p = 1$, $\alpha = 0.8$ for ARP-PPO (scaled).
- InvertedDoublePendulum-v2: $p = 1$, $\alpha = 0.5$ for ARP-PPO and $p = 1$, $\alpha = 0.95$ for ARP-PPO (scaled).
- InvertedPendulum-v2: $p = 1$, $\alpha = 0.3$ for ARP-PPO and $p = 1$, $\alpha = 0.95$ for ARP-PPO (scaled).
- Swimmer-v2: $p = 1$, $\alpha = 0.95$ for ARP-PPO and $p = 3$, $\alpha = 0.8$ for ARP-PPO (scaled).
- Reacher-v2: $p = 1$, $\alpha = 0.3$ for ARP-PPO and $p = 3$, $\alpha = 0.8$ for ARP-PPO (scaled).
- Hopper-v2: $p = 3$, $\alpha = 0.5$ for ARP-PPO and $p = 1$, $\alpha = 0.95$ for ARP-PPO (scaled).
- Walker2d-v2: $p = 1$, $\alpha = 0.5$ for ARP-PPO and $p = 1$, $\alpha = 0.8$ for ARP-PPO (scaled).

Figure 14 shows the training curves with respect to both physical time and training time (details in Appendix F.2). It is observed that SAR outperforms ARPs often by a large margin on both criteria.

## H Results with $\lambda$-SAR

To verify whether $\lambda$-SAR described in Section 4.4 could be effective in partially observable MDPs (POMDPs), we modify the stochastic MuJoCo environments with the "Strong External Force (Perceptible)" setting (described in Section 5.2) by adding partial observability. Specifically, in addition to the stochasticity, we make the environment cause a penalty reward of $r_{\text{penalty}}$ when the agent holds the same action more than or equal to $t_{\text{thres}}$ seconds, where the agent cannot observe the current holding time of an action, which renders the environment to be a POMDP. We test methods on the four MuJoCo environments of InvertedPendulum-v2, InvertedDoublePendulum-v2, Hopper-v2 and Walker2d-v2 with the lowest $\delta$'s, where we set $t_{\text{thres}} = 0.04$ and $r_{\text{penalty}} = -1$ for InvertedPendulum-v2, $t_{\text{thres}} = 0.04$ and $r_{\text{penalty}} = -10$ for InvertedDoublePendulum-v2, and $t_{\text{thres}} = 0.025$ and $r_{\text{penalty}} = -20$ for Hopper-v2 and Walker2d-v2. Figure 15 shows the training curves of $\lambda$-SAR-PPO ($\lambda = 0.5$), SAR-PPO and FiGAR-C-PPO. We confirm that $\lambda$-SAR can cope with such partial observability by incorporating temporal information into safe regions.

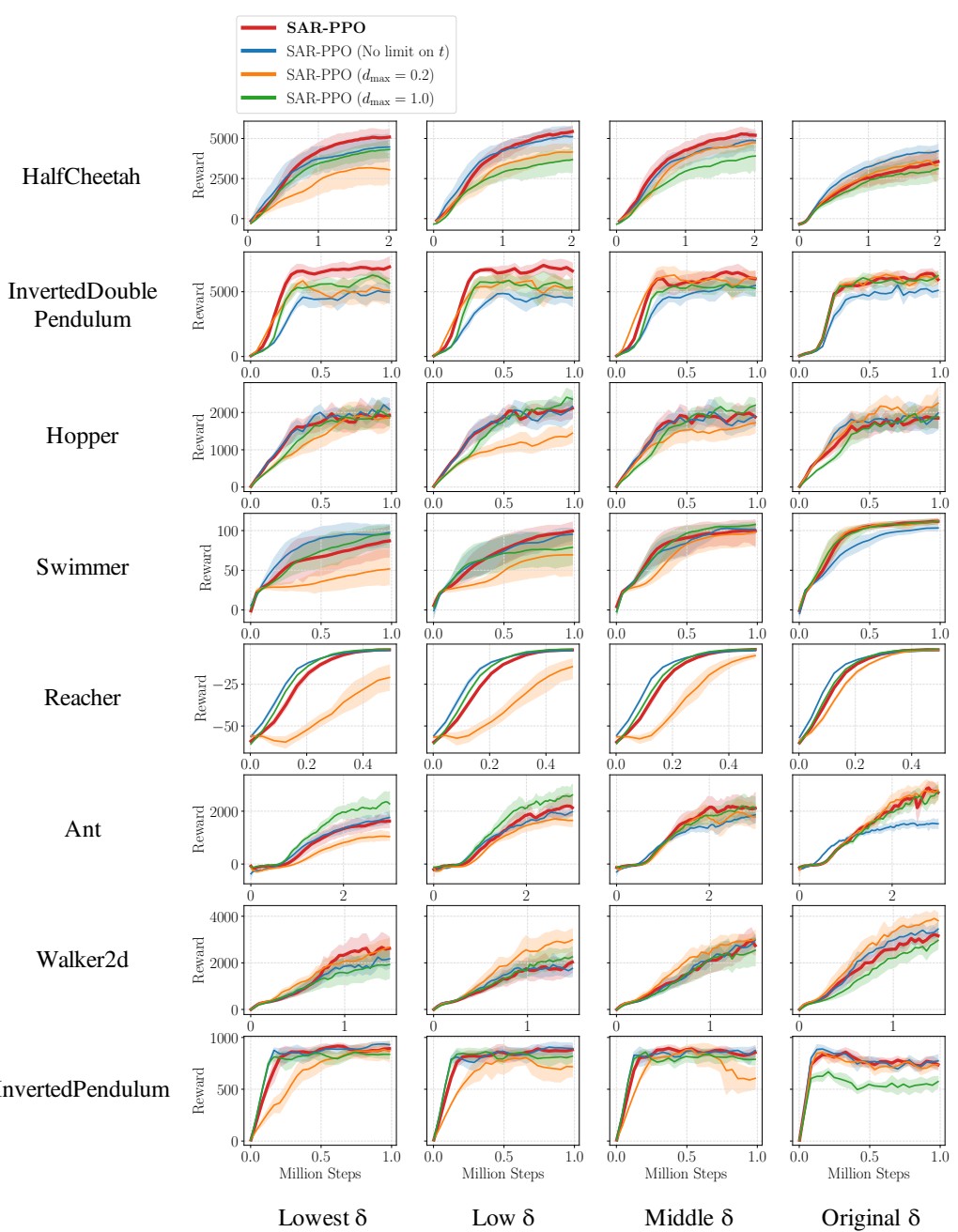

Figure 16: Training curves of multiple variations of SAR-PPO on eight deterministic MuJoCo environments with various $\delta$'s. Shaded areas represent the $95\%$ confidence intervals over eight runs.

# I  Ablation Study

**Variants of SAR-PPO.** We test SAR-PPO with its variations. As stated in Appendix J, we fix $d_{max} = 0.5$ in SAR for the experiments in the main paper. We alter $d_{max}$ to $0.2$ ("SAR-PPO ($d_{max} = 0.2$)") or $1.0$ ("SAR-PPO ($d_{max} = 1.0$)") to demonstrate how this hyperparameter affects the performance of SAR. We also experiment with another variant of SAR ("SAR-PPO (No limit on $t$)") that does not impose an upper limit on the maximum duration of actions. Figure 16 shows the results on eight deterministic MuJoCo environments. We observe that a small $d_{max}$ may lead

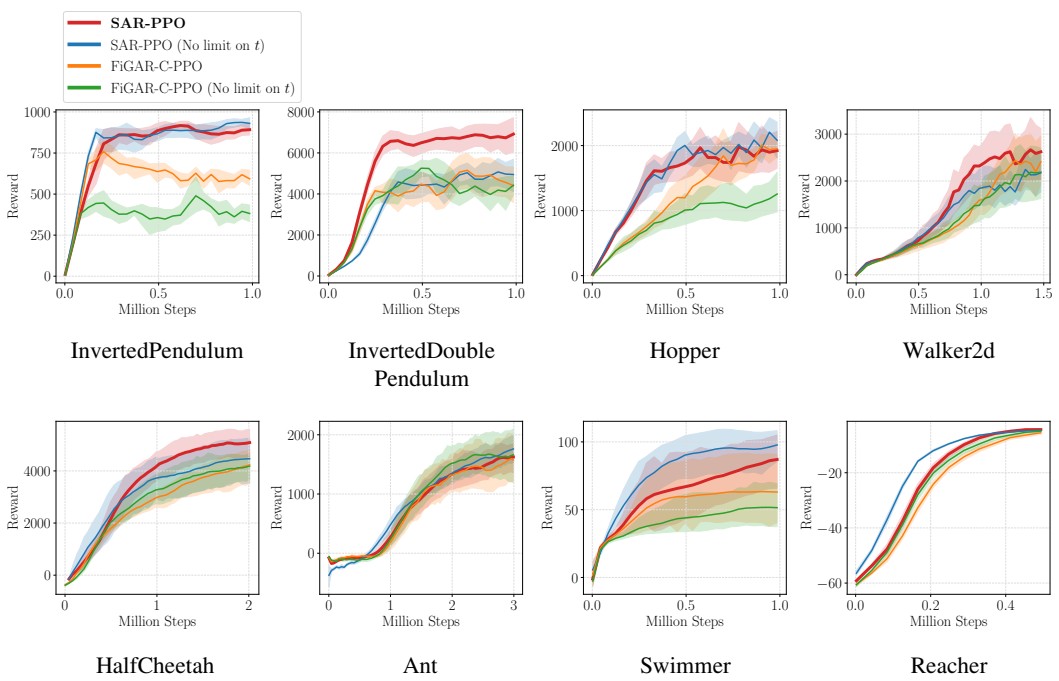

Figure 17: Training curves of SAR-PPO, FiGAR-C-PPO and their variants without $t$ limit on eight deterministic MuJoCo environments with the lowest-$\delta$ settings. Shaded areas represent the $95\%$ confidence intervals over eight runs.

to inferior performance on average since it may excessively limit action durations, increasing the average number of decision steps and thus the variance of the PG estimator.

**Effect of a limit on $t$.** In order to examine the effect of imposing an upper limit on action durations, we test variants of SAR-PPO and FiGAR-C-PPO. "SAR-PPO (No limit on $t$)" denotes the same setting as the previous experiment and "FiGAR-C-PPO (No limit on $t$)" denotes the setting of FiGAR-C-PPO without clipping $t$ while it uses the same scale of $t$ as our original FiGAR-C-PPO. Figure 17 demonstrates that in both setting, imposing a limit on $t$ leads to better performance on most of the environments as it helps stabilize training, although SAR-PPO (No limit on $t$) sometimes outperforms the original SAR-PPO on some environments such as Reacher-v2.

**Variants of SAR-PPO's distance function.** We use the $\ell_1$ norm for the distance function of SAR: $\Delta(s, s_i) = \|\tilde{s} - \tilde{s_i}\|_1 / \dim(\mathcal{S})$. In this experiment, we test another variant of SAR with the $\ell_2$ norm, whose distance function is defined as $\Delta(s, s_i) = \|\tilde{s} - \tilde{s_i}\|_2 / \sqrt{\dim(\mathcal{S})}$. We set $d_{\max} = 0.5$ for the $\ell_1$ norm and $d_{\max} = 1.0$ for the $\ell_2$ norm. Figure 18 suggests that the $\ell_1$ norm is slightly more effective than the $\ell_2$ norm. We speculate that this is because some state dimensions with large changes may dominate $\ell_2$ distances.

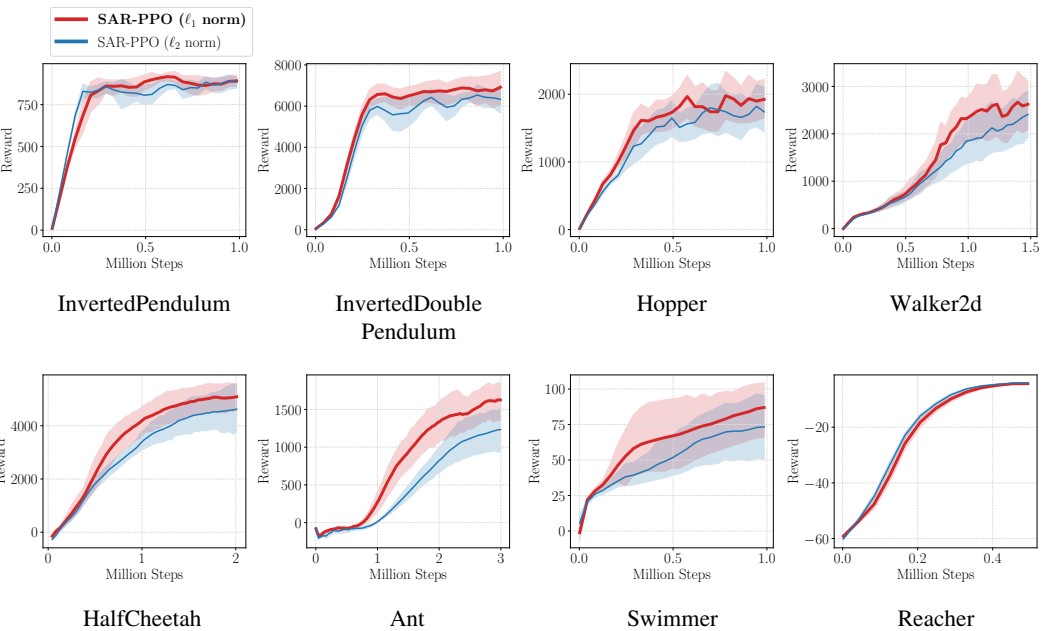

Figure 18: Training curves of SAR-PPO with the $\ell_1$ or $\ell_2$ norm on eight deterministic MuJoCo environments with the lowest-$\delta$ settings. Shaded areas represent the $95\%$ confidence intervals over eight runs.

Table 1: Discretization time scales.

| Environment | $\delta_{\text{lowest}}$ | $\delta_{\text{low}}$ | $\delta_{\text{middle}}$ | $\delta_{\text{original}}$ ($\delta_0$) |
|---|---|---|---|---|
| Ant-v2 | $2e-3$ | $5e-3$ | $1e-2$ | $5e-2$ |
| HalfCheetah-v2 | $2e-3$ | $5e-3$ | $1e-2$ | $5e-2$ |
| InvertedDoublePendulum-v2 | $2e-3$ | $5e-3$ | $1e-2$ | $5e-2$ |
| InvertedPendulum-v2 | $2e-3$ | $5e-3$ | $1e-2$ | $4e-2$ |
| Swimmer-v2 | $2e-3$ | $5e-3$ | $1e-2$ | $4e-2$ |
| Reacher-v2 | $1e-3$ | $2e-3$ | $5e-3$ | $2e-2$ |
| Hopper-v2 | $5e-4$ | $1e-3$ | $2e-3$ | $8e-3$ |
| Walker2d-v2 | $5e-4$ | $1e-3$ | $2e-3$ | $8e-3$ |

## J   Experimental Details

### J.1   Implementation

We implement SAR and the baseline methods based on the open-source implementations of Stable Baselines3 [25] (a port of Stable Baselines [12] for PyTorch[24]) for PPO [30] and A2C [20], and Stable Baselines [12] for TRPO [28]. We use the publicly released official implementations for DAU [36] (`https://github.com/ctallec/continuous-rl`) and ARP [14] (`https://github.com/kindredresearch/arp`). We provide the implementation for our experiments (including licenses) in the anonymous repository at `https://vision.snu.ac.kr/projects/sar`.

### J.2   Environments

We experiment on eight continuous control environments from MuJoCo [37]: InvertedPendulum-v2, InvertedDoublePendulum-v2, Hopper-v2, Walker2d-v2, HalfCheetah-v2, Ant-v2, Reacher-v2 and Swimmer-v2.

The environment parameters used in our experiments are as follows:

- Ant-v2: $\mathcal{S} = \mathbb{R}^{111}$, $\mathcal{A} = [-1, 1]^8$, $\sigma_{\text{act}} = 1$, $p_{\text{act}} = p_{\text{ext}} = 0.05$, $\sigma_{\text{ext}} = 100$, $\sigma_{\text{ext2}} = 300$.

- HalfCheetah-v2: $\mathcal{S} = \mathbb{R}^{17}$, $\mathcal{A} = [-1, 1]^6$, $\sigma_{\text{act}} = 1$, $p_{\text{act}} = p_{\text{ext}} = 0.05$, $\sigma_{\text{ext}} = 30$, $\sigma_{\text{ext2}} = 300$.

- InvertedDoublePendulum-v2: $\mathcal{S} = \mathbb{R}^{11}$, $\mathcal{A} = [-1, 1]^1$, $\sigma_{\text{act}} = 1$, $p_{\text{act}} = p_{\text{ext}} = 0.05$, $\sigma_{\text{ext}} = 100$, $\sigma_{\text{ext2}} = 1000$.

- InvertedPendulum-v2: $\mathcal{S} = \mathbb{R}^4$, $\mathcal{A} = [-3, 3]^1$, $\sigma_{\text{act}} = 3$, $p_{\text{act}} = p_{\text{ext}} = 0.05$, $\sigma_{\text{ext}} = 300$, $\sigma_{\text{ext2}} = 1000$.

- Swimmer-v2: $\mathcal{S} = \mathbb{R}^8$, $\mathcal{A} = [-1, 1]^2$, $\sigma_{\text{act}} = 1$, $p_{\text{act}} = p_{\text{ext}} = 0.05$, $\sigma_{\text{ext}} = 100$, $\sigma_{\text{ext2}} = 1000$.

- Reacher-v2: $\mathcal{S} = \mathbb{R}^{11}$, $\mathcal{A} = [-1, 1]^2$, $\sigma_{\text{act}} = 1$, $p_{\text{act}} = p_{\text{ext}} = 0.05$, $\sigma_{\text{ext}} = 300$, $\sigma_{\text{ext2}} = 1000$. Due to its unique environment dynamics, we apply external torques instead of forces.

- Hopper-v2: $\mathcal{S} = \mathbb{R}^{11}$, $\mathcal{A} = [-1, 1]^3$, $\sigma_{\text{act}} = 1$, $p_{\text{act}} = p_{\text{ext}} = 0.05$, $\sigma_{\text{ext}} = 30$, $\sigma_{\text{ext2}} = 300$.

- Walker2d-v2: $\mathcal{S} = \mathbb{R}^{17}$, $\mathcal{A} = [-1, 1]^6$, $\sigma_{\text{act}} = 1$, $p_{\text{act}} = p_{\text{ext}} = 0.05$, $\sigma_{\text{ext}} = 100$, $\sigma_{\text{ext2}} = 1000$.

For the discretization time scales, we generally follow the values from Tallec et al. [36]. Table 1 shows the values we used for $\delta$'s. We use an episode horizon of 1000 and a discount factor of $\gamma_0 = 0.99$ for all the environments with the original discretization time scale ($\delta_0$). In lower-$\delta$ settings, we scale the episode length by $\delta_0/\delta$ to maintain the same physical time limit, and set the discount factor to $\gamma_0^{\delta/\delta_0}$ to have the same effective horizon. We discount the reward both between decision steps (accordingly to Equation (5)) and during action repetitions in SAR and FiGAR-C. For the "Strong External Force (Perceptible)" setting described in Section 5.2, we append to the state the applied 3-D force (or 3-D torque in the case of Reacher-v2) vector clipped to a range of $[-1, 1]$.

Table 2: Hyperparameters for PPO.

| Hyperparameter | Value |
|---|---|
| Optimizer | Adam |
| Learning rate | $1e - 4$ |
| Nonlinearity | ReLU |
| # decision steps per train step | 2048 |
| # epochs per train step | 10 |
| Minibatch size | 64 |
| GAE parameter $\lambda$ | 0.95 |
| Clipping parameter $\epsilon$ | 0.2 |

Table 3: Hyperparameters for TRPO.

| Hyperparameter | Value |
|---|---|
| Optimizer | Adam |
| Learning rate | $1e - 4$ |
| Nonlinearity | ReLU |
| # decision steps per train step | 1024 |
| GAE parameter $\lambda$ | 0.95 |
| KL step size | 0.01 |
| Conjugate gradient damping factor | 0.1 |
| # iterations for conjugate gradient | 10 |
| # iterations for the value function | 5 |
| Minibatch size for the value function | 128 |

Table 4: Hyperparameters for A2C.

| Hyperparameter | Value |
|---|---|
| Optimizer | RMSProp |
| Adam learning rate | $1e - 4$ |
| Nonlinearity | ReLU |
| # decision steps per train step | 256 |

## J.3 Training

Throughout the experiments, we model each learnable component with an MLP with two hidden layers of 256 dimensions. For the policies of PPO, TRPO and A2C, we use a normal distribution with a learnable diagonal covariance matrix that is independent of states, following the implementations of Schulman et al. [28, 30]. For the policies of SAR and FiGAR-C, we modify the variance corresponding to $d$ or $t$ actions to be dependent on states. We normalize returns (rewards) and each dimension of states using their moving averages for the inputs of the components in all environments except Ant-v2; we find that it performs better not to use the normalization in Ant-v2. For SAR's distance function, we use normalized states in all environments in order to make safe regions agnostic to the scale of each state dimension. We set $d_{\max} = 0.5$ (chosen among $\{0.1, 0.2, 0.5, 1.0\}$) for SAR and $t_{\max} = 0.05$ (chosen among $\{0.01, 0.02, 0.05, 0.1\}$) for FiGAR-C, and share them across all the environments. We run our experiments on our internal CPU cluster mostly consisting of Intel Xeon E5-2695 v4 and Intel Xeon Gold 6130 processors. Each run in our experiments usually takes 3-12 hours on a single CPU core.

We report the hyperparameters used for each RL algorithm in Tables 2 to 4. For further implementation details, we refer to our released code as well as the official implementations of DAU and ARP.