# OpenReview forum: "Time Discretization-Invariant Safe Action Repetition for Policy Gradient Methods"
_NeurIPS.cc/2021/Conference — NeurIPS 2021 Poster_

### Official Review · Reviewer_CWkP · 2021-07-14

**Rating:** 6
**Confidence:** 4

**Summary:**

This work is motivated by the performance dropoff of RL algorithms when the time scale $\delta$ decreases to 0. Although previous work has studied such an issue in Q-learning,  this paper rather focuses on policy-gradient (PG). Observing that the variance of PG explodes as $\delta\rightarrow 0$, the authors introduce Safe Action Repetition (SAR), which ensures bounded gradient regardless of the time scale $\delta$. Inspired by action repetition, the proposed technique is additionally able to handle unexpected events by interrupting repetition when the agent reaches an unsafe/uncertain region in the state-space.

**Limitations And Societal Impact:**

See details above

**Main Review:**

*Originality*

This work brings $\delta$-robustness forward by incorporating it into policy-gradient methods. To my knowledge, the idea behind SAR is new for tackling varying time scales. By accounting for spatial structure, SAR adds a new criterion for action repetition.

*Quality and significance*

I raise a few questions and concerns below:
1. In Thm 1, why is the dependence on $\sigma_1$ only? I would expect dependence on the whole covariance matrix, or on some norm of it. Is there an implicit assumption according to which the variance coefficients in $\Sigma$ are ordered? If so, please mention it.

2. l. 74 - "this is the first action repetition method that repeats an action based on the agent's state" and criterion in Eq. (6) therein: It is not clear to me how this distance threshold accounts for safety. Implicitly, it means that the furthest the agent is away from the reference state, the least safe it is. What is the motivation for such an approach? Why not repeat actions based on reward variation instead? For example, given some reference state and action (with corresponding reference reward), the agent could perform action repetition until the reward drops down. It would not be based on a norm because of the asymmetry of that criterion (since higher reward is desirable so you would repeat the action if doing so goes in your favor), but seems like an equally valid measure.

3. In Proposition 2, several parameters vary at once, all of them tending to an extreme case ($\delta,x\rightarrow 0, \nu\rightarrow \infty$ and $f=0$). As such, it is not clear how each of these parameters leads to variance explosion. What would happen if $\delta \rightarrow 0$ much slower than $\nu\rightarrow \infty$? All of these interdependencies may affect the behavior of the variance.

4. How does SAR evolve as the stochasticity level increases? In particular, I guess this would result in smaller safe regions and consequently reduce action duration. This brings me to considering Eq (8), which introduces a trade-off between distance and time: one would probably have to calibrate $\lambda$ accordingly maybe by setting $lambda \rightarrow 1$ as stochasticity increases. Such analysis is missing in the experiments.

5. The first paragraph in Sec 6 reminds the setting of options, which should be addressed in related work:  Why not consider an option approach instead of action repetition? What would be missing there compared to SAR?

*Significance*

Overall, the problem is interesting and the proposed solution numerically promising, but the paper does not sufficiently challenge its method against other alternatives, which questions how much other researchers would later use SAR.  The limitations of SAR are hardly mentioned (only in Conclusion when suggesting an extension to POMDPs).


*Clarity*

Although the ideas are clearly described, there are several English mistakes that I enumerate below.
Additionally, below are some points that remain unclear to me:
1. l. 283-284 "Note that this variant of SAR.. is $\delta$-invariant too" - proof? Justification?
2. l. 279 "$\lambda$ is the coefficient that controls the relative strength of the first term" - it rather combines the first term with the second one by trading off distance with time difference.
3. l. 296 "While other previous ..." - I did not understand the point of this sentence
4. l. 351-353 "... SAR can quickly and adaptively handle such stochasticity with safe regions by immediately stopping repetitions" - How is this interpreted from the plot?



*Typos and minor comments*

- Redundancies in l. 1 - "discretized", "discretization"; l. 33 - "differently", "different";
- "the" should be removed in the following: l. 37 - "the performance"; l. 38 - "the trained policies"; l. 49 - "the access"; l. 56 - "the prior"; l. 93 - "the collapse"; l. 189 - "the training"; l. 190 - "the computational cost"; l. 215-216 - "the state s, not the time t"; l. 218 - "the environments"; l. 231 - "the repetition"; l. 232 - "the action repetition"; l. 238 - "the action repetition"; l. 257 - "the time"; l. 267 - "the variance explosion"; l. 268 - "the optimal policy"; l. 271 - "the stochasticity"; l. 273 - "the variance explosion"; l. 4 Fig 3 caption - "the full results"; l. 282 - "the time at $s_i$", "the repetition"; l. 304 - "the training"; l. 305 - "the duration"; l. 2 Fig 4 caption - "the $95%$"; l. 342 - "the position", "the velocity"; l. 364 - "the locality"
- l. 18 "assume an MDP" --> "are based on an MDP" or "rely on an MDP" (MDP is not an assumption but a model)
- l. 21 "the MDP assumption" --> "the MDP setting" or "the MDP model"
- l. 20 "are defined with continuous-time" --> "in continuous time" (note hyphen removal as well)
-  l. 21 "the continuous-time" --> "the continuous timeline"
- l. 40 "There have been proposed some methods" --> "(Some) methods have been proposed" ("Some" is superfluous here also)
- l. 45 "its agility" --> "agility"
- l. 44 "when it is necessary" --> "when necessary"
- l. 47 "by preventing the collapse of the Q function" --> "by preventing from Q-function collapse"
- l. 59 "failures" --> "failure"
- l. 63 "the be not only $\delta$-invariant" --> "to not only be $\delta$-invariant"
- l. 72 "any PG methods" --> "any PG method" or "all PG methods"
- l. 95 "extended ... with" --> "extended ... to"
- l. 100 "prevent the variance explosion" --> "prevent from variance explosion"
- l.121 + l. 205 "given as"--> "given by"
- l. 138 "a solution for" --> "a solution to"
- l. 193 "but does only when it is necessary" --> "but does so when necessary only"
- l. 206 "it is too applicable" --> "it is also applicable"
- l. 209 "replacing ... with" --> "replace ... by"
- l. 210 "times of repetition" --> "repetition times"
- l. 212 "durations of actions" --> "action duration"
- l. 217 "caring underlying changes" --> "caring about underlying changes"; "degradation in performance" --> "performance degradation"
- l. 215 "for a fully observable MDP" --> "of a fully ...."
- l. 218 "vary greatly" --> "greatly vary"
- l. 219 "we will demonstrate" --> "we demonstrate"
- l. 231 "outside the safe region" --> "outside of the safe region"
- l. 234 "since its action duration, which is determined by the radius of a safe region, has no relation with how fine the discretization time scale is" --> "since action duration is determined by the safe region radius, which is not related to how fine the discretization timescale is"
- l. 236 "and how it is scaled" --> "and the way it is scaled"
- l. 259 "good num actions to maximize f(num)" --> "num actions that maximize f(num)"
- l. 260 "the optimal policy" --> "an optimal policy" (there may be more than one); "for off that" --> "for off such that"
- l. 261 "for num that" --> "for num such that"
- l. 266 "the full proof in Appendix" --> "a full proof in the Appendix"
- l. 269 "it can be also at risk of failure" --> "it can also be a risk of failure"
- l. 275 "be finitely bounded" --> "be bounded" (if it is were infinite then it would not be bounded)
- l. 277 "we derived SAR based on that the optimal policy ... depends only on states" --> "We derive SAR based on the fact that an optimal policy ... only depends on states"
- l. 4 Fig 3 caption "We refer to Appendix" --> "to the Appendix"
- l. 287 "We first demonstrate $\delta$-invariance of" --> "the $\delta$-invariance of"
- l. 282 "repetition. $0 \leq \lambda \leq 1$" --> "repetition,  $0 \leq \lambda \leq 1$"
- l. 296 "While other previous ... approaches" --> "Although previous ... approaches"
- l. 308 "though they have randomized" --> "although they have randomized"
- l. 288 "comparing with" --> "compared to"
- l. 313 "in Appendix" --> "in the Appendix"
- l. 314 "their performances" --> "their performance"
- l. 358 "with the stochasticity of environments" --> "with stochasticity" or "with environment stochasticity"
- l. 364 "the locality of states" --> "states locality"

POST REBUTTAL: I am happy to give an additional point to the authors for their clarifications.



**Time Spent Reviewing:**

4 hours

---

> ### Author Response · Authors · 2021-08-10
> **Author Response to Reviewer CWkP**
>
> We sincerely appreciate the reviewer’s valuable and insightful feedback.
>
> **1. Dependence only on $\sigma_1$**
>
> Since we use $\sigma_1$ without loss of generality (L528 in Appendix B), we can replace $\sigma_1$ with any other $\sigma_k$ where $k \in \\{1, 2, \ldots, K\\}$, or more generally $\text{min}(\sigma_k)$, and $\sigma$'s are not particularly ordered.
> In Theorem 1, the choice of the subscript does not matter because our goal is to only prove that the RHS of Equation (4) explodes to infinity when $\delta \to 0$.
> However, in order to prevent potential confusion, we will clarify this in the final draft by replacing $\sigma_1$ with $\text{min}(\sigma_k)$.
>
> **2. Motivation for repeating actions based on the agent's states**
>
> The motivation for SAR is based on the implicit assumption that the optimal policy $\pi^*(a|s)$ would output similar actions for similar states. Shen et al. [1] also empirically validate this assumption. On the other hand, previous approaches such as FiGAR-C are implicitly based on the assumption that the optimal actions are similar within a short time range; however, it is not supported by the form of the optimal policy $\pi^*(a|s)$, which is not related to $t$. As such, it can lead to performance degradation where states change much in a short period of time (L217). Therefore, the optimal behavior can be more *safely* approximated by repeating the same action within a specific region around the *state* not *time* (L215 - L224).
>
> The suggested reward-based approach could be an alternative, but it may not work well in sparse reward environments where rewards are given with a long delay, or in constant reward environments such as InvertedPendulum-v2 where a fixed reward is generated at each step. This approach may not decide when to stop actions in these environments. Moreover, since the reward $R(s)$ is often a function of states in most RL settings, states contain richer information than rewards.
>
> [1] Qianli Shen, Yan Li, Haoming Jiang, Zhaoran Wang, and Tuo Zhao. Deep reinforcement learning with robust and smooth policy. ICML 2020.
>
> **3. Parameters’ effects on Proposition 2**
>
> For Proposition 2, we have $\delta$, $x$, $\nu$, $f$ as environment parameters.
> First, $\nu \to \infty$ is just a technical realization of a very large penalty (e.g., $- \nu = - \infty$) given when the action $\mathrm{off} = 1$ is not taken upon an alert. We set $f=0$ to simplify the analytical derivation. For $x$ and $\delta$, $x > \delta > 0$ always holds (L252). Consequently, $x \to 0$ is the *only* extreme case we need to consider. Intuitively, it is an example of how SAR and FiGAR-C would learn in the environment where the agent should turn the switch off once it is alerted and the grace period ($x$) goes to $0$.
>
> How it leads to variance explosion: FiGAR-C has no choice but to inﬁnitely shorten action durations in order to immediately turn off the alarm (because the grace period tends to 0), while SAR can instantly react to the change in the alarm state utilizing safe regions even if $x \to 0$.
>
> We will make the explanation more intuitive in the final draft.
>
> **4. How does SAR evolve as the stochasticity level increases?**
>
> First, FiGAR-C (i.e., $\lambda = 0$ in $\lambda$-SAR) should reduce action durations when the stochasticity level increases, in order to quickly respond to unexpected events. FiGAR-C is unaware of underlying state changes, and thus its best strategy is to shorten the duration of actions to be more responsive.
> On the other hand, SAR (i.e., $\lambda = 1$ in $\lambda$-SAR) does not necessarily shrink the size of safe regions even if the stochasticity level increases. This is advantageous as a short action duration can cause the variance explosion problem (Theorem 1).
> The intuition is that SAR is already responsive since it can stop the current action immediately after the state deviates from the safe region. SAR can simply set the safe region size enough to *detect* the presence of unexpected events, and the size is not necessarily related to the frequency of unexpected events.
>
> We additionally perform an experiment to empirically illustrate this on InvertedPendulum-v2 ($\delta = 2e-3$) with external forces. We test $\lambda \in \\{0.0, 0.1, 0.3, 1.0\\}$ and $p_\text{ext} \in \\{0.025, 0.05, 0.1, 0.2\\}$ where $p_\text{ext}$ means the probability of the external force being applied. We train $\lambda$-SAR with PPO for 1M steps and compute the result with 8 random seeds for each setting.
>
> | $p_\text{ext}$ | Avg. normalized radius ($\lambda=0$, FiGAR-C) | Avg. normalized radius ($\lambda=0.1$) | Avg. normalized radius ($\lambda=0.3$) | Avg. normalized radius ($\lambda=1$, SAR) |
> | -: | -: | -: | -: | -: |
> | 0.025 | **0.94** | **0.94** | **0.95** | **0.89** |
> | 0.05 | 0.78 | **0.91** | **0.92** | **0.93** |
> | 0.1 | 0.35 | 0.76 | **0.96** | **0.91** |
> | 0.2 | 0.15 | 0.50 | 0.83 | **0.93** |
>
> | $p_\text{ext}$ | Avg. reward ($\lambda=0$, FiGAR-C) | Avg. reward ($\lambda=0.1$) | Avg. reward ($\lambda=0.3$) | Avg. reward ($\lambda=1$, SAR) |
> | -: | -: | -: | -: | -: |
> | 0.025 | 170 | 387 | 819 | 847 |
> | 0.05 | 92 | 256 | 765 | 889 |
> | 0.1 | 55 | 146 | 607 | 929 |
> | 0.2 | 55 | 75 | 447 | 869 |
>
> The first table shows the learned (normalized) sizes of safe regions and the second table reports the average rewards. The learned radius reduces as the stochasticity level increases in FiGAR-C ($\lambda=0$), while such shrinkage does not happen in SAR ($\lambda=1$). The result also suggests that the original SAR outperforms FiGAR-C and $\lambda$-SAR with $0 < \lambda < 1$. Note that since InvertedPendulum-v2 is a fully observable MDP, $\lambda$-SAR, which is designed for POMDP, does not outperform the original SAR in this environment (as a side note, one can find the result where $\lambda$-SAR performs the best in Appendix H).
>
> **5. Why not consider an option approach instead of action repetition?**
>
> As the review suggests, the options framework and action repetition are related since they both use temporally extended actions. However, our goal is to propose a *$\delta$-invariant* policy gradient algorithm, which is different from the goal of the options framework. Another crucial difference between them is that within the options framework, the agent has to produce each $\delta$-discretized low-level action (even with open-loop options), which makes having $\delta$-invariance non-trivial.
>
> In addition, with the closed-loop options, the number of decision steps increases as $\delta \to 0$, which makes its low-level policy (i.e., intra-option policy) suffer from the variance explosion problem.
>
> We will include further discussion on the similarity and differences between action repetition and the options framework (as well as semi-Markov decision processes) and cite relevant work in the final paper.
>
> **6. The paper does not sufficiently challenge its method against other alternatives**
>
> We appreciate the comment but respectfully disagree with the reviewer on this point.
> In our submission, on various settings of base RL algorithms and stochasticity types, we compared our method with many diverse approaches for $\delta$-invariance or low $\delta$’s:
>
> (1) FiGAR-C (our modification of FiGAR for $\delta$-invariance),
>
> (2) DAU (a $\delta$-invariant approach for Q-learning),
>
> (3) the scaled PPO variant in Appendix F (our modification of PPO to reduce the PG variance by adjusting parameters as in DAU), and
>
> (4) ARPs in Appendix G (a PG method that uses an autoregressive process to prevent the variance explosion problem).
>
> Also, we ablated components in SAR in Appendix I for a better understanding of our method, and we will additionally include the result on variants of SAR’s distance functions (please see our response to Q2 of Reviewer hLjy).
>
> **7. The limitations of SAR are hardly mentioned**
>
> The limitations of SAR were mentioned in Conclusion (about partially observable settings and our distance metric), Section 4.4 (about partially observable settings) and Appendix A (about potential negative societal impact). We will make them more clear and noticeable. Also, as our response to Reviewer hLjy, QGiy and HCbD, we will move the Broader Impact in Appendix A to the main paper, and further clarify the limitations on our distance metrics and the assumption of Theorem 1 in the final draft.
>
> **8. Clarity**
>
> - "Note that this variant of SAR.. is $\delta$-invariant too" - proof? Justification?
>
> Since Equation (8) can be viewed as a weighted combination of SAR and FiGAR-C where each of them is $\delta$-invariant (L211, L234), $\lambda$-SAR is $\delta$-invariant too. We will clarify this in the final version.
>
> - "$\lambda$ is the coefficient that controls the relative strength of the first term" - it rather combines the first term with the second one by trading off distance with time difference.
>
> We agree with your clarification. We will update our draft following your suggestion.
>
> - "While other previous ..." - I did not understand the point of this sentence
>
> We will clarify in the final draft that we made the comparison despite its unfairness to SAR due to the requirements of those baselines (ARPs (Appendix G) and DAU (Section 5.1)), such as expensive computational cost.
>
> - "... SAR can quickly and adaptively handle such stochasticity with safe regions by immediately stopping repetitions" - How is this interpreted from the plot?
>
> “quickly” and “immediately”: In Figure 6, SAR stops repeating actions immediately (only taking one or two steps - which can be seen from the number of markers between the *break point* and the point where the external force is applied) after the stochastic events.
>
> “adaptively”: SAR produces the safe regions with different sizes according to the current velocity (L347).
>
> **9. Typos and minor comments**
>
> We deeply appreciate your sincere efforts to improve the presentation quality of our paper.
> In the final version, we will happily update and clarify all the points that the review suggested.

---

### Official Review · Reviewer_HCbD · 2021-07-16

**Rating:** 7
**Confidence:** 3

**Summary:**

Real world tasks are often in continuous time, whereas typical RL algorithms operate in discrete time. In such situations the algorithms are often sensitive to the chosen discretization time step.

Small discretization scales can lead to various problems such as increases in the gradient variance, high computational costs, and difficulties with exploration.

This paper proposes safe action repetition (SAR) to tackle these issues. SAR augments the action space with an additional variable that determines for how long the chosen action will be repeated. In particular, this additional variable will determine the size of an L1 ball around the current state, such that when the agent leaves this ball, a new action will be chosen, i.e. it determines the size of a "safe region" during which the action will be repeated.

One previous related work that tackled this problem was FiGAR that used a similar augmentated action, but which determined the number of action repeats, not the size of a safe region. A downside of this approach is that it can not react when something goes wrong during the action repeat period.

They perform experiments on MuJoCo continuous control tasks combining their algorithm with PPO, TRPO and A2C and show invariance to variance time discretization lengths and improved performance. They compared to FiGAR as well as DAU, another algorithm for discretization invariance (they compared to the official implementation). Their method usually considerably outperformed the other algorithms.

They also proved the explosion of the policy gradient variance when the discretization goes to 0 under the assumption that the variance of the return stays lower bounded.

------------------------------------------------------------------------
Update: See the comments for more discussion. The score remained unchanged.

**Ethical Concerns:**

No concern

**Limitations And Societal Impact:**

Yes

**Main Review:**

In general the paper was well written, the tackled problem relevant and the approach sensible. The experiments included ablation studies. They also included small toy examples explaining the benefits of their method. One thing that was still unclear to me was why the agent would simply not learn to set the safe region as small as possible to maximize the control frequency and the achievable performance? I hope the authors can explain this in their response. I also point out that the assumption that the variance of the return is lower bounded may not hold.

**Variance may not be lower bounded**

Even if the discretized environment is stochastic, as the discretization time step is reduced, the effect of the stochasticity can become smaller and the outcome can become deterministic in the continuous limit. For example, this will happen in the example by Munos that you cited, and actually, if one takes multiple samples and uses a mean baseline, the policy gradient variance would not explode in their example. One could also construct other examples where the return variance keeps decreasing as the discretization time scale is reduced. Based on this, I believe some of the claims are a bit too strong, and it should be emphasized that it relies on the assumption that the variance is lower bounded, which does not hold in general (though there are many examples where indeed the PG variance will explode, and the experiments in the appendix also demonstrated this empirically.)

**Comment on proposition 2**

The example is OK, but I found it somewhat artificial to just place noise on the final reward to ensure that reward variance stays lower bounded. Another point that bothered me was that the choice of the size of the safe region has no effect on the task performance because there are only two states 0 and 1, and irrespective of the size of the safe region, the next action will be chosen right as the state changes. On the other hand, if the state were to change continuously, wouldn't the policy with $d_i \to 0$ also be optimal, and lead to an infinite amount action steps and an explosion of the gradient variance also for SAR (this is basically related to my previous question of why the agent does not just learn to set the safe region as small as possible)?

**Time Spent Reviewing:**

4h

---

> ### Author Response · Authors · 2021-08-10
> **Author Response to Reviewer HCbD**
>
> We deeply appreciate the reviewer’s insightful and constructive feedback.
>
> **1. Why won't SAR simply set safe regions as small as possible?**
>
> The radius $d$ decreases when the gain from fine-grained control with smaller $d$’s is greater than that with larger $d$’s by a non-negligible margin. Otherwise, policy gradient signals would not guide $d$ to be smaller, since a low $d$ leads to an increase in the PG estimator's variance (i.e., signals would *not* be *consistent enough*). Moreover, gradient signals for SAR often favor a larger $d$ for more exploration; a high $d$ usually leads to better coverage of the state space (e.g., Figure 1), especially in (but not limited to) the early stages of the training, and thus the probability of encountering a state with a high expected return by chance is higher with a larger $d$.
>
> **2. Variance may not be lower bounded**
>
> As the review points out, it is not always possible to lower-bound the return variance by some positive constant, which leads to the necessity of the assumption for Theorem 1 (L167 and L535). We also acknowledge that increasing the batch size ("if one takes multiple samples") or decreasing the learning rate can reduce the variance (L176).
>
> Nonetheless, we believe that our assumption often holds in practice even if the reward function is not stochastic. Intuitively, the assumption states that the return variance is greater than some $c > 0$ given any fixed random seed for action sampling, where random seeds correspond to the reparameterized actions $\epsilon_{0:N-1}$. Environments with deterministic transition dynamics and reward functions, such as MuJoCo environments, could also satisfy this assumption, considering the stochasticity of the initial state.
>
> However, following the suggestion, we will revise the expressions by clearly stating the implications and necessity of the underlying assumption in the final version.
>
> **3. On Proposition 2**
>
> We let the reward function be stochastic and the state space be discrete in order to make the analytic upper and lower bounds of the PG estimator's variance more tractable. However, we acknowledge that the illustrative example might be somewhat artificial.
>
> If we consider this example in the continuous state space, SAR with a radius of $d=0.5$ still can attain the optimal policy without experiencing variance explosion. At the same time, it is true that multiple optimal solutions for SAR exist even with small values of $d$ (e.g., $d=0.00001$). However, we do not expect that the learned radius would be particularly small given any reasonable initialization of the parameters (due to the reason in our response to Q1). On the other hand, FiGAR-C achieves the optimality only when its action duration $t$ becomes infinitesimal, as it is unaware of when the state changes.
>
> In order to empirically illustrate that FiGAR-C reduces action durations while SAR does not in the presence of stochasticity, we perform an additional experiment on InvertedPendulum-v2 ($\delta = 2e-3$) with varying stochasticity levels of external forces. We train SAR and FiGAR-C with $p_\text{ext} \in \\{0.025, 0.05, 0.1, 0.2\\}$ where $p_\text{ext}$ means the probability of the external force being applied. For each setting, we train the agent with PPO for 1M steps using 8 random seeds.
>
> | $p_\text{ext}$ | Avg. normalized action duration (FiGAR-C) | Avg. normalized radius (SAR)|
> | -: | -: | -: |
> | 0.025 | **0.94** | **0.89** |
> | 0.05 | 0.78 | **0.93** |
> | 0.1 | 0.35 | **0.91** |
> | 0.2 | 0.15 | **0.93** |
>
> The table above shows learned (normalized) action durations and the radiuses of safe regions with various levels of stochasticity. While the action duration of FiGAR-C reduces as the stochasticity level increases, the radius of SAR does not.
>
> For the more detailed results, please see our response to Q4 of Reviewer CWkP.

---

> > ### Comment · Reviewer_HCbD · 2021-09-07
> > **More comments**
> >
> > Thanks for the response; it mostly answered my questions, and I am keeping my score.
> >
> > The justifications for why SAR won't simply set the safe region as small as possible seem reasonable but "hacky", and I think that finding a more principled method/explanation to ensure that the safe region won't go to 0 would be a good topic for future work. But it works empirically, so I think it's fine for now.
> >
> > Regarding the variance lower bound, it is fine if you add clear statements. However, I think the assumption is a bit stronger, as you require that $c\not\to 0$ as $N\to \infty$. For example, in a feedback control system, increasing $N$ will make the controller more responsive and better at responding to noise, so even if the system dynamics are stochastic, the trajectory could become deterministic in the limit $N\to\infty$. Moreover, my comment about "multiple samples" was about using the samples to construct a baseline, which you stated that you are ignoring. In practice, one should always use a baseline together with policy gradient methods. When including a baseline, your analysis should be modified to look at the difference of the return from the baseline. In such a situation, when considering deterministic dynamics, even if you consider a stochastic initial state, a perfect baseline could cancel out this stochasticity. However, in practice the baseline may not be perfect.

---

> > > ### Author Response · Authors · 2021-09-07
> > > **Author Response to Reviewer HCbD**
> > >
> > > Thanks for the additional feedback!
> > >
> > > First, as the review mentions, since the perfect baseline may not exist in practice,
> > > we expect that our assumption could hold (at least empirically) in many situations including the MuJoCo environments, which can explain the failure of vanilla PG algorithms in our experiments.
> > >
> > > Nevertheless, we agree with your comment, especially in that
> > > a perfect baseline can eliminate the variance
> > > in environments with deterministic dynamics and a stochastic initial state
> > > (although it could not when the dynamics or rewards are also stochastic),
> > > and there can exist some stochastic tasks (with either deterministic initial states or perfect baselines) where trajectories could become deterministic when $N \to \infty$,
> > > which necessitates the assumption.
> > >
> > > We appreciate your active engagement in the discussion and the helpful comments.
> > > We will update our draft to contain the mentioned aspects (including the PG baseline).

---

### Official Review · Reviewer_QGiy · 2021-07-18

**Rating:** 7
**Confidence:** 5

**Summary:**

This paper studies the effect of discretization of the time on the policy space methods in RL. Since the underlying physics of the most of the RL systems is in continuous time, one needs to perform a discretization of the time horizon to employ policy gradient to find the optimal policy. However, as shown in the paper, policy space methods might suffer from high variance when the discretization parameter is small. In addition, a large discretization parameter is not also optimal, because it will result in a suboptimal policy, and it is not robust to stochastic behavior of the system. To deal with this problem, the authors propose SAR (safe action repetition). The idea is to let the algorithm repeat the same set of actions for a window of time, until the system exists some "safe region". To find this safe region, the policy outputs the radius of the safe region, and the algorithm needs to find the optimal policy which maximizes the reward while finding the optimal safe region. The authors compare their algorithm with the previous work experimentally, and they show that their algorithms outperforms FiGAR-C.

**Limitations And Societal Impact:**

Need for notion of distance in the state space

**Main Review:**

Overall, it is a good work. However, I have some questions:
- What is V(s) in equation (5)?
- You used the notion of a distance in the state space to create the safe region. How can we extend this method to the systems where one cannot define a notion of distance on the state space?
- In SAR, the radius d_i is one of the outputs of the policy and needs to be learned while we learn the optimal policy. What if the initialization of the policy assigns very small (or very large) radiuses in some states, and hence the algorithm will be divergent? In other word, I am concerned about the effect of the initialization of the policy (specially the safe region radius d_i) on the final policy achieved by the algorithm. How can we make the algorithm less sensitive to the initialization of the policy?
- Although the authors provide figure 6 which shows that SAR produces reasonable safe regions, I was wondering how is the algorithm encouraged to find the best safe region? In other words, I want to know, in an intuitive sense, how the algorithm is rewarded to end up getting a policy which gets a larger safe region when the speed is low and bigger safe region when the speed is high?


**Time Spent Reviewing:**

12

---

> ### Author Response · Authors · 2021-08-10
> **Author Response to Reviewer QGiy**
>
> We deeply appreciate the reviewer’s constructive and thoughtful feedback.
>
> **1. $V(s)$ in Equation (5)**
>
> $V(s)$ is the value function (L130). It is the output of the value network for $s$ as input.
>
> **2. Necessity of the distance function defined in the state space**
>
> As the review points out, SAR operates in environments where the distance functions can be properly defined in the state spaces. There may be some cases where it is nontrivial to define the distance function such as discrete or very high dimensional state spaces. In such cases, we can perform SAR in the learned feature space of the states [1, 2] (Section 6) or using learned distance metrics [3], which can be intriguing future work.
> We will further clarify this point in the final draft.
>
> [1] Danijar Hafner, Timothy Lillicrap, Jimmy Ba, and Mohammad Norouzi. Dream to control: Learning behaviors by latent imagination. ICLR 2020.
>
> [2] Hao Liu, Pieter Abbeel. Behavior from the void: Unsupervised active pre-training. ArXiv, abs/2103.04551, 2021.
>
> [3] Robert Dadashi, Shideh Rezaeifar, Nino Vieillard, Léonard Hussenot, Olivier Pietquin, Matthieu Geist. Offline Reinforcement Learning with Pseudometric Learning. ICML 2021.
>
> **3. Effect of the initialization of the policy**
>
> If the policy is initialized within a rational range, SAR will learn appropriate sizes of safe regions. Figure 11 in Appendix E illustrates this. It shows how average action durations increase or decrease over time in SAR and FiGAR-C (where we plot $t$ instead of $d$ to illustrate both algorithms in the same figure).
>
> SAR may not work well if the initial value of $d$ is excessively small or large. However, it is easily avoidable as we use the distance function of the $\ell_1$ norm on *normalized* states divided by the number of state dimensions (L299). This makes the same initialization scheme applicable to many environments including MuJoCo tasks.
>
> **4. How is the algorithm encouraged to find the best safe region?**
>
> In InvertedPendulum-v2 (Figure 6), the agent receives a constant reward at each $\delta$-discretized step. When the agent's velocity is low, a bigger safe region is a low-hanging fruit especially in the early stages of the training where the PG estimator is mostly dominated by immediate rewards (i.e., accumulated reward within a single repetition).
> On the other hand, when the velocity is high and if safe regions are too large, the pendulum would easily lose the balance and thus lead to the end of the episode, which makes SAR in favor of smaller safe regions.
>
> For how the safe region expands or shrinks in general, please see our response to Q1 of Reviewer HCbD.

---

### Official Review · Reviewer_hLjy · 2021-07-19

**Rating:** 6
**Confidence:** 4

**Summary:**

The paper proposes an alternative to durative actions where instead of outputting an action and how long the action is to persist, it instead outputs an action and a radius. This radius defines a "safe region" where the action is to be repeated for as long as an agent is within it. Having the action duration depend on state makes it invariant to the environment's time discretization. They evaluate their proposal on the MuJoCo suite and demonstrate improved performance over an agent which bases its durative action on time alone.

**Limitations And Societal Impact:**

Adequately addressed, but put in the supplementary material. I believe it is a requirement for it to be in the main text, as per the following post by the program chairs:

"Whereas the broader impacts section previously did not count towards the page limit, it must now fit within the page limit. However, we have extended the page limit from 8 pages to 9 pages..."

**Main Review:**

I think the paper is overall well written and nicely organized. The motivating environment was really good for building intuition around key limitations of time-based action repetition, as well as the merit of a feature-based one.

I'm recommending weak acceptance at this time, as I believe it meets the bar for publication, but feel it missed an important comparison for painting a clearer empirical understanding of the proposed method:

Could the authors comment on how imposing t_max on SAR ensures a fair comparison between the two methods? For two orthogonal approaches for controlling action repetition, I think this instead blurs the comparison in that the paper lacks a clear/isolated evaluation of the proposed method. From the suggestion that imposing this limit stabilizes things, it would be informative to know the degree at which things are unstable without it, and some explanation for why it fails (e.g., are there notable degenerate cases that arise?). I think the results are promising in that the combination of the two appears to perform well, but think that such an omission detracts from a clear empirical understanding of the proposed method. Similarly, though more distant from this paper's scope, one could instead consider removing t_max from FiGAR-C to understand the role of this limit, e.g., is it a bias to make results more consistent or easier to tune, or is it *necessary* for getting the algorithm to work?

I additionally have the following question, which largely didn't affect my rating: Have the authors tested alternative distance metrics? Can they comment on the sensitivity of the approach to such a choice, or give intuitions around how it might be chosen given a problem (e.g., correlated features in the observation, etc.)?

**Time Spent Reviewing:**

6

---

> ### Author Response · Authors · 2021-08-10
> **Author Response to Reviewer hLjy**
>
> We deeply appreciate the reviewer’s thoughtful and constructive feedback.
>
> **1. Imposing $t_{\text{max}}$**
>
> We imposed $t_{\text{max}}$ on SAR for two reasons: fair comparison and stability. We used the *same* $t_{\text{max}}$ on both SAR and FiGAR-C for a fair comparison to make sure that they have the identical expressiveness of action repetition in terms of duration; without it, SAR can have a very long duration that FiGAR-C cannot. With regard to stability, $t_{\text{max}}$ can help SAR avoid some failure modes such as outputting overly large safe regions due to inappropriate initialization.
>
> To analyze the effect of $t_{\text{max}}$ on SAR, we performed an ablation study without $t_{\text{max}}$ in Figure 17 in Appendix I. It demonstrates that SAR shows similar performance even without $t_{\text{max}}$ in most cases, although SAR attains improved performance with it in a few environments, such as InvertedDoublePendulum-v2.
>
> As per the reviewer's suggestion, we additionally experiment FiGAR-C without $t_{\text{max}}$ on the eight deterministic MuJoCo environments with the lowest $\delta$’s. For each setting, we train the agent with PPO for 1M steps using 8 random seeds.
>
> | Environment | Avg. reward (SAR w/ $t_{\text{max}}$) | Avg. reward (SAR w/o $t_{\text{max}}$) | Avg. reward (FiGAR-C w/ $t_{\text{max}}$) | Avg. reward (FiGAR-C w/o $t_{\text{max}}$) |
> |:-|-:|-:|-:|-:|
> | Ant-v2 |276.02 |**481.55** |183.88 |237.66 |
> | HalfCheetah-v2  |**4195.96** |3735.57 |2976.01 |3282.08 |
> | Hopper-v2 |1872.59 |**2116.1**  |1919.2  |1293.25 |
> | InvertedDoublePendulum-v2  |**7016.85** |4938.03 |4329.32 |4284.69 |
> | InvertedPendulum-v2  |899.51 |**933.38** |575.32 |398.03 |
> | Reacher-v2  |**-3.95** |-4.3  |-4.19 |-4.18 |
> | Swimmer-v2  |87.38 |**98.89** |63.13 |51.38 |
> | Walker2d-v2 |**2401.96** |1846.85 |1622.41 |1475.51 |
>
> The table above shows that SAR outperforms FiGAR-C regardless of the presence of $t_{\text{max}}$. The results also suggest that the role of $t_{\text{max}}$ on FiGAR-C is not very significant in most environments, but it helps stabilize FiGAR-C in some environments like Hopper-v2 and InvertedPendulum-v2. We will include the result and further discussion in the final draft.
>
> **2. Alternative distance metrics**
>
> We additionally test SAR with the normalized $\ell_2$ distance as another distance metric on the eight MuJoCo environments with the lowest $\delta$’s. We use the $\ell_2$ variant with $d_{\text{max}} = 1$ and train the agent with PPO for 1M steps using 8 random seeds.
>
> | Environment | Avg. reward (SAR with $\ell_1$) | Avg. reward (SAR with $\ell_2$) |
> |:-|-:|-:|
> | Ant-v2 |276.02 |13.26 |
> | Ant-v2 (at 3M steps) |1624.80 |1234.92 |
> | HalfCheetah-v2  |4195.96 |3397.97 |
> | Hopper-v2 |1872.59 |1735.99 |
> | InvertedDoublePendulum-v2  |7016.85 |6442.64 |
> | InvertedPendulum-v2  |899.51 |885.15 |
> | Reacher-v2  |-3.95 |-3.72 |
> | Swimmer-v2  |87.38 |73.23 |
> | Walker2d-v2 |2401.96 |1810.5  |
>
> SAR with the $\ell_1$ distance slightly outperforms SAR with the $\ell_2$ distance. Presumably, it is because some state dimensions with large changes may dominate $\ell_2$ distances. As the performance difference between them is marginal in most environments, we expect that SAR may not be sensitive to the choice of the distance metric in many cases (also note that the MuJoCo environments contain some correlated features).
>
> In very high-dimensional environments, it would be interesting to consider distance learning or representation learning techniques (Section 6).
>
> **3. Limitations and societal impact**
>
> The [NeurIPS FAQ](https://neurips.cc/Conferences/2021/PaperInformation/NeurIPS-FAQ) mentions that "... may include a discussion of these potential negative societal impacts anywhere in the paper (in the intro, in the conclusion, as a stand-alone section, *in the supplemental material* if appropriate, etc.)". However, since the final revision allows one more extra page, we will move the section into the main paper.

---

### Decision · Program_Chairs · 2021-09-27

**Decision:**

Accept (Poster)

**Comment:**

This paper proposes a novel trick to address the time discretization issue with continuous-time policy gradient method. As a small time interval delta is used, the authors shows that variance of PG estimator can exploded when delta->0. To address the issue, they propose a Safe Action Repetition method, which lets the action repeat when the state changes within a small ball with diameter d. This approach guarantees safe and robust policy optimization that is delta-invariant and adaptive to exploration spped. All reviewers appreciate that this solution is interesting and potentially useful.

The paper is still improvable. For example, the delta-invariance is achieved by reducing the time discretization problem to identifying a small neighborhood of the state space. This technique heavily depends on a priorly available state distance metric. Finding a good state distance metric is nontrivial, and would require further research beyond this paper.